# Evaluating the Anti-Corruption Factor in Environmental, Social, and Governance Indices by Sampling Large Financial Asset Management Firms

Kenneth David Strang [1,2,3,*] and Narasimha Rao Vajjhala [4]

1  Business Analytics Department, W3-Research, St. Thomas, VI 00802, USA
2  Plaster Graduate School of Business, University of the Cumberlands, Williamsburg, KY 40769, USA
3  Business School, RMIT University, Melbourne, VIC 3004, Australia
4  Faculty of Engineering and Architecture, University of New York Tirana, 1048 Tirana, Albania; narasimharao@unyt.edu.al
*  Correspondence: professor@kennethstrang.com or kenneth.strang@gmail.com

**Abstract:** Current Environmental, Social, and Governance (ESG) indices are flawed because the data are incomplete and not reported consistently, and some measured factors may be irrelevant to the industry. Regulators in the financial services industry emphasize reporting $CO_2$ emissions (environmental factor), yet the key resources leveraged for production are rented offices, and internet–governance issues like money laundering, corruption, and unethical behavior would be more relevant. To investigate this problem, we sampled the finance and insurance industry firms in the USA with the greatest economic impact, i.e., those managing at least USD 1 trillion in assets. We used artificial intelligence to collect data about undisclosed legal decisions against firms to measure the ESG anti-corruption governance factor GRI 206-1, defined by the Global Reporting Institute (GRI) for global sustainable development goals (SDGs), which correspond to the United Nations' SDGs. We applied Bayesian correlation with bootstrapping to test our hypotheses, followed by root cause analysis. We found that ESG ratings from providers did not reflect legal cases decided against firms; the Bayesian $BF_{+0}$ odds ratio was 3005 (99% confidence intervals were 0.617, 0.965). Also, misconduct fines and arbitration legal case counts were significantly related for the same firm (the Vovk-Selke maximum p-ratio was 4411), but most ESG scores were significantly different for the same firm. We found three other studies in the literature that corroborated some of our findings that specific firms in our sample were considered to be unethical. We propose deeper study of the implications related to our findings based on public interest and stakeholder theory.

**Keywords:** ESG; sustainability; financial transparency; governance; asset management; SEC; FINRA; NASAA; unethical behavior; misconduct; disclosure; expungement; stakeholder theory; public interest theory

## 1. Introduction

The United Nations (UN)'s 17 sustainable development goals (SDGs) are designed to measure global sustainability [1]. SDG compliance is an important topic to study.

### 1.1. Rationale for Research Problem

However, more than one researcher has reported difficulty in analyzing contributions at the organizational level [2–8]. For instance, it is impractical to measure the reduction in human trafficking or the increase in clean water availability when your firm is operating in a rented New York City office. The SDGs are important for future sustainability, and since organizations have the greatest impact on ecology, it makes sense to study how firms can improve SDG compliance.

Several associations have proposed firm-level constructs to improve the measurement of global sustainability progress. James Stewart at the University of Sydney, Australia, collaborated with the UN to develop the Environmental, Social, and Governance (ESG) model, which has become the de facto scorecard for organizations [1]. Still, many researchers have complained that the ESG model does not accurately measure sustainability progress, and they have accused some firms of promoting ESG alignment in their marketing brochures without substantiating any credible evidence of organic ESG contributions [3,4,9]. Researchers claim that there is a high degree of inconsistency between ESG scores for the same firms, which lowers the ESG model's credibility, reliability, and validity [10,11].

Greenwashing is the term commonly used to describe unethical ESG behavior [12,13]. For example, one firm in the financial services industry appointed a content writer, gave them an executive title, tasked them with creating online narratives encouraging employees to recycle plastic water bottles, and then claimed that this was an ESG contribution. While it was a nice gesture, good and bad quantitative ESG outputs need to be disclosed. Dieselgate is a classic example of greenwashing. Volkswagen marketed their car as "clean diesel" but manipulated the automotive computer to cover up the true levels of emissions; this unethical behavior caught up with them when they subsequently reported a loss of EUR 1.7 billion [14]. BMW, Mercedes, and others were found guilty of similar unethical greenwashing behavior [12,14].

There have been numerous studies about ESG quantification and greenwashing in the financial industry. Blackrock (https://www.blackrock.com/, accessed on 24 May 2024) is considered to be one of the largest financial asset management firms in the world, with over USD 10.8 trillion in assets under management (AUM). Blackrock has come under scrutiny from researchers about their ESG claims. For example, Mrchkovska, Dolšak, and Prakash [15] and Etchart [16] found that Blackrock's CEO created media narratives about reducing environmental $CO_2$ emissions while avoiding the social and governance factors. There may have been a legitimate reason for this, because financial industry regulators are emphasizing reporting the environmental factor in the ESG model. Blackrock stated in a recent 10-K filing (https://s24.q4cdn.com/856567660/files/doc_financials/2024/ar/BLK_AR23.pdf, p. 13, accessed on 24 May 2024), under the ESG section, that the U.S. SEC regulator will require them to make substantial climate-related disclosures with respect to governance and greenhouse gas emissions. There were no ESG-related governance disclosures except under the 'legal proceedings' caption, stating that "from time to time, BlackRock receives subpoenas or other requests for information from various . . . regulatory authorities . . . (and that) certain of its subsidiaries and employees have been named as defendants in various legal actions, including arbitrations and other litigation" (p. 27).

*1.2. Research Questions*

This issue is perplexing, because it is questionable how a financial services firm could greatly impact environmental sustainability in terms of greenhouse gas emissions, since they typically rent office buildings and use the internet for their operations. It would be much different if Blackrock were a farming or gas refinery company. A financial services firm ought to have a greater impact on governance factors such as ethics, money laundering, and anti-corruption. The extent, quantity, and nature of Blackrock's legal proceedings were not disclosed, which would presumably be part of their governance in the ESG framework. This raises an interesting dilemma: do adversarial legal proceedings, including regulatory fines, impact ESG and SDG sustainability?

The answer is yes, according to Global Reporting Institute (GRI), whereby legal proceedings are mentioned under the anti-corruption category within the governance factor [17]. GRI has provided two measurable items to quantify anti-corruption governance, i.e., (a) the number of legal actions pending or completed during the reporting period regarding anti-competitive behavior and violations of anti-trust and monopoly legislation in which the organization has been identified as a participant; and (b) the main outcomes

of completed legal actions, including any decisions or judgments [17]. If the answers are not zero, then each increment above zero would lower the firm's governance rating.

Based on the above financial industry problems and issues, the research questions (RQs) underpinning the current study are:

- RQ1: Do common ESG sustainability indexes of large financial asset management firms with AUM > $1 trillion accurately measure anti-corruption governance?
- RQ2: Are large financial asset management firms disclosing anti-corruption governance to comply with GRI 206-1 (number and outcomes of legal actions)?

### 1.3. How ESG Is Related to SDG Sustainability Measurement

The 17 SDGs adopted by all United Nations Member States in 2015 provide a blueprint for peace and prosperity, addressing global challenges, including poverty, inequality, climate change, environmental degradation, peace, and justice [18]. The Environmental (E) component of ESG addresses climate change, resource use, waste management, and environmental conservation [19]. The "E" component directly aligns with SDGs such as SDG 6—Clean Water and Sanitation, SDG 7—Affordable and Clean Energy, SDG 13—Climate Action, SDG 14—Life Below Water, and SDG 15—Life on Land [19]. The Social (S) aspect focuses on a company's relationships with its employees, suppliers, customers, and communities, covering labor practices, human rights, and community engagement, aligning with SDG 1—No Poverty, SDG 3—Good Health and Well-being, SDG 4—Quality Education, SDG 5—Gender Equality, SDG 8—Decent Work and Economic Growth, and SDG 10—Reduced Inequality [20].

This study focuses on the governance (G) component, which evaluates a company's leadership, executive pay, audits, internal controls, and shareholder rights [21]. Good governance is fundamental for achieving the SDGs, especially SDG 16—Peace, Justice, and Strong Institutions and SDG 17—Partnerships for the Goals [21]. Governance plays a critical role in the ESG framework. Good governance practices establish the foundation for ethical business conduct, ensuring accountability, fairness, and transparency in a company's stakeholder relationship [21]. Effective governance helps identify, manage, and mitigate financial, legal, and reputational risks. This is essential for long-term sustainability and resilience. Strong governance practices are crucial for aligning corporate strategies with the SDGs, as these practices ensure that a company's actions contribute positively to global challenges, such as reducing corruption (SDG 16) and fostering partnerships (SDG 17) [22]. Good governance ensures that companies comply with relevant laws and regulations, including those related to environmental protection and social equity, which is necessary for sustainable business operations and contributes to achieving the SDGs [22]. Governance practices emphasizing transparency, accountability, and ethical behavior help build and maintain trust with stakeholders, including investors, employees, customers, and the community [23]. Governance in ESG reporting refers to the mechanisms, processes, and relations by which corporations are controlled and directed, including board composition and structure, for instance, the diversity and independence of board members [24]. This component also includes executive compensation, dealing with the alignment of compensation with long-term performance, shareholder rights, protecting shareholder interests, and equitable treatment, and ethical conduct, dealing with the policies on ethics, anti-corruption, and compliance [24].

Governance is directly linked to several SDGs focusing on institutional integrity and partnerships, including SDG 16—Peace, Justice, and Strong Institutions and SDG 17—Partnerships for the Goals [25]. SDG 16 has two essential aspects: anti-corruption and accountability [25]. Governance practices that enforce anti-corruption measures are vital for achieving SDG 16, and companies also need to implement strong internal controls, conduct regular audits, and ensure transparency to prevent corrupt practices [26]. Also, companies need to perform reporting on governance practices to enhance transparency and accountability, which are crucial for building strong institutions. SDG 17 has two essential aspects—collaboration and policy coherence, as effective governance facilitates collaboration with

various stakeholders, including governments, NGOs, and other corporations, to achieve the SDGs, and governance practices ensure that a company's policies are aligned with national and international sustainability goals [27].

ESG criteria have become integral in evaluating investment sustainability and ethical impact, particularly for financial asset management firms with assets under management (AUM) exceeding $1 trillion. However, the reliability of these ESG ratings in accurately reflecting anti-corruption and human rights governance in alignment with the SDGs is under scrutiny. The ESG ratings provided by agencies such as MSCI, Sustainalytics, and FTSE Russell aim to offer comprehensive assessments of a company's sustainability practices [28]. However, studies indicate substantial discrepancies in these ratings, particularly concerning anti-corruption and human rights governance. Berg et al. [29] found significant differences in ESG ratings for the same firms across various providers. These discrepancies are attributed to methodological differences, criteria selection, and weightings [29]. For instance, MSCI may prioritize corporate governance metrics differently than Sustainalytics, leading to varying assessments of the same firm's practices [29]. Such inconsistencies undermine the reliability of ESG ratings in accurately measuring anti-corruption and human rights governance. Chatterji et al. [30] highlighted that the lack of standardization in ESG rating methodologies results in varied interpretations of a company's sustainability performance, particularly in areas like anti-corruption policies and human rights practices.

Anti-corruption governance is critical for achieving SDG 16, which promotes peace, justice, and strong institutions, and this can be achieved by effective anti-corruption measures involving comprehensive policies, risk assessments, and transparent reporting. According to Hess [31], many firms disclose anti-corruption policies without providing detailed information on their implementation and effectiveness. This lack of depth in reporting compromises the accuracy of ESG ratings in reflecting actual anti-corruption practices. Human rights governance, integral to SDGs 8 and 10, involves ensuring that corporate operations address labor rights, non-discrimination, and community impacts. Eccles and Stroehle [32] found that while firms frequently report on human rights policies, they lack detailed reporting on their implementation and impact. This gap in reporting raises questions about the reliability of ESG ratings in accurately reflecting human rights governance. The inconsistencies in ESG ratings and the lack of detailed reporting on anti-corruption and human rights governance support the hypothesis that popular ESG ratings do not reliably measure these practices. The significant differences in ratings for the same firms further underscore this unreliability.

*1.4. Measuring SDG and ESG Anti-Corruption Governance Using GRI*

Integrating ESG criteria into investment decision-making has gained significance among large financial asset management companies [33]. These criteria are used to evaluate the sustainability and ethical impact of an investment in a company or business [34]. However, the accuracy of ESG indexes in terms of measuring anti-corruption and human rights governance, particularly in alignment with the best practices for the Sustainable Development Goals (SDGs), remains a topic of scrutiny [35,36]. ESG sustainability indexes, such as the Dow Jones Sustainability Index (DJSI), FTSE4Good, and the MSCI ESG Index, serve as benchmarks for investors seeking to integrate sustainability into their portfolios [28,37].

ESG sustainability indexes assess companies based on various criteria across environmental, social, and governance dimensions, including climate change policies, labor practices, corporate governance, and ethical standards [38]. Anti-corruption governance refers to companies' policies, procedures, and actions to prevent and combat corruption within their operations and supply chains [39]. This is critical to corporate governance and sustainability, as bribery can undermine trust, economic development, and social stability. SDG 16 (Peace, Justice, and Strong Institutions) emphasizes the importance of solid anti-corruption measures [40].

Studies have shown that while ESG indexes include anti-corruption metrics, there are significant variations in how these metrics are defined, measured, and weighted [41–43].

For instance, the DJSI assesses anti-corruption based on corporate policies, risk assessments, and disclosure practices but does not uniformly verify the implementation and effectiveness of these measures [44,45]. Also, many ESG indexes rely heavily on self-reported data from companies, which can be biased or incomplete [46,47].

One major criticism is the lack of standardization across ESG indexes, because different indexes may use varying criteria and weightings for anti-corruption governance, leading to inconsistent and sometimes contradictory assessments of the same company [48–50]. The reliance on self-reported data without independent verification raises concerns about the accuracy and reliability of the information used in these indexes [46]. However, Kolk [50] underlines the crucial need for more rigorous and transparent methodologies. This emphasis on methodological robustness is key to improving the credibility of ESG ratings. The above literature questions if the ESG rating providers provide credible, reliable and valid scores. We propose the following hypothesis to test if popular ESG rating providers calculate reliable, valid and credible scores:

**H1:** *Popular ESG ratings do not consistently measure anti-corruption governance because they differ significantly for the same financial asset management firms with AUM > $1 trillion.*

Human rights governance ensures that a company's operations respect and promote human rights throughout its value chain [51]. This includes labor rights, non-discrimination, health and safety, and community impacts. SDG 8 (Decent Work and Economic Growth) and SDG 10 (Reduced Inequality) underscore the importance of human rights in sustainable development [52,53]. ESG indexes typically include human rights criteria, but the depth and breadth of these criteria can vary [34]. The FTSE4Good Index, for example, assesses human rights performance based on policies, management systems, and disclosure practices [54]. However, as with anti-corruption metrics, there is often a reliance on self-reported data [55].

Eccles and Stroehle [32] found that while ESG indexes have progressed in incorporating human rights metrics, there are still significant gaps in coverage and consistency. For example, issues such as forced labor, child labor, and fair wages are not uniformly assessed across all indexes [32]. Moreover, the effectiveness of corporate policies in addressing these issues is often not thoroughly evaluated. The primary limitations of ESG indexes in measuring human rights governance stem from the lack of standardized criteria and the reliance on self-reporting [56]. Furthermore, there is often insufficient focus on the outcomes of corporate policies and practices.

The SDGs provide a comprehensive framework for sustainable development, with specific targets and indicators for measuring progress. Best practices for anti-corruption and human rights governance under the SDGs involve having robust policies in place and demonstrating effective implementation and positive outcomes. Research indicates that while ESG indexes increasingly align their criteria with the SDGs, significant challenges remain [57–59]. The GRI has provided the substance for measuring governance. As explained earlier, GRI recommends disclosing the number of adversarial legal proceedings where a firm was named a participant and reporting the outcomes [17]. The GRI asks for a list or counts of completed legal proceedings. We propose the following hypothesis to test if the largest financial asset management firms meet the GRI 206-1 anti-corruption governance:

**H2:** *The number of adversarial legal outcomes (misconduct, unethical behavior, regulatory violations) of large financial asset management firms is not accurately reflected in popular ESG scores.*

## 2. Methods and Materials

### 2.1. Research Design and Methodology

The authors applied a post-positivist ideology, and the strategy to address the RQs was to collect and analyze quantitative data to test the hypotheses using a 99% confidence level. A positivist ideology is appropriate when the variables of interest are numeric (or could be

transformed to numeric values) and robust parametric statistical techniques are available to test hypotheses [60].

The analysis was at the firm level. There was no dependent variable. Although the RQs and hypotheses implied comparing ESG scores from different providers, this was not a between-group strategy; instead, it was a within-group design. The objective was to determine how strongly the firm's ESG scores from several providers were related to one another, as they should be if they are measuring the same firm data. Correlation methods are appropriate to estimate the strength and direction of relationships in the data for a within-group strategy when no dependent variable exists [61]. If, instead, we were comparing the ESG scores between firms, that would be a between-group strategy, where ANOVA-type methods could be used. Another objective was to determine if the anti-corruption human rights governance items were accurately counted and reflected in the ESG scores. Descriptive and correlation statistical methods could be used to test those types of hypotheses [61].

To test the first hypothesis (H1: Popular ESG ratings do not consistently measure anti-corruption governance because they differ significantly for the same financial asset management firms with AUM > $1 trillion), Pearson and Bayesian correlation were applied with bootstrapping. The assumptions for Pearson's Product Moment estimate were met, because the ESG scores were continuous variable data types (even though they were mostly interval), the ESG values were collected from independent providers identified from other studies in the literature (a priori), and the population was sampled based on a filter independent of the tested values, i.e., AUM > $1 trillion. The ESG scores were sampled from unrelated sources. The Shapiro-Wilk statistic was utilized to ensure the ESG scores had multivariate normal distributions.

Bootstrapping was used, which increases the sample size 1000 times while maintaining the same data distribution. Bootstrapping can provide more power for coefficient estimates of quantitative data when the sample size is small [62], allowing us to use Bayesian correlation and Pearson to test the hypotheses. Although we applied Spearman correlation as a method of triangulation, we did not report those coefficients due to space limitations if they did not change the interpretation of the results. The Vovk-Selke maximum p-ratio was calculated, in addition to the standard *p*-value tests. It is a robust hypothesis test, primarily when bootstrapping is used with a small sample size containing numeric data types. This test helps avoid accepting false positives and negatives in hypothesis testing [62]. We also needed a robust correlation coefficient and effect size estimate to overcome missing data values, because several ESG indexes did not have ratings for some firms.

Bayesian correlation was applied to interpret the correlation hypothesis test between ESG scores. This test furnishes an odds ratio to indicate the strength of the evidence supporting the hypotheses being tested, reducing the likelihood of accepting false positives or negatives [63]. Two critical estimates are called Bayes Factors (BFs). According to Wetzels and Wagenmakers [63], the BF01 estimate represents the statistical odds that the null side of the hypothesis is true (no correlation) compared to the data under the alternative side. The second Bayes Factor coefficient is BF10, which is the opposite of BF01, indicating the odds that the alternative side of the hypothesis is true (strong correlation, positive or negative) based on the data. A BF10 > 1 is desired [63], analogous to a *p*-value < 0.01 when the confidence level is 99%. The BF10 odds ratio allows the researcher to infer how likely the evidence supports the alternative hypothesis. For example, a BF10 = 2 implies the alternative hypothesis is twice as likely to be true based on the data. We cannot make such inferential claims with traditional correlation coefficients. Although Bayesian correlation can further strengthen the model by using prior and posterior probabilities, those ESG probabilities were unavailable to the authors. Our algorithm also calculated a kappa coefficient between 0 and 2 to estimate the likelihood the results were not found by chance—this was important in a small sample with lower data ranges. Credibility intervals were calculated to support inferential generalization.

### 2.2. Population and Sampling

The population was the financial asset management industry globally, with a sample drawn from large firms in the USA with over $1 trillion of assets under management (AUM). The largest firms were selected because more data were available for their compliance with SDG best practices, and the implications of the current study would have a broader impact if the sampled population consisted of those with more assets and more customers. In other words, the sample represents an important economic component of the national gross domestic product. It is important that researchers accurately describe the intended population and that the sample accurately represents the population so that the study results may be considered reliable, valid, and credible to support inferential generalizations [60].

We adopted the widely used NAICS (https://www.census.gov/naics/, accessed on 25 May 2024), managed by the U.S. government, to identify the population and sample frame. The NAICS code used for population identification was 523940, which covers firms that provide financial planning, portfolio management, and investment advice, and 523150–523160, companies that provide brokerage services of financial investment products. Our sampling technique to develop the sample frame was purposive, which filtered data to match the RQ and hypothesis criteria. We reduced the sample frame to those firms with 10-K form filings in the previous year and having ESG ratings. Descriptive statistics were used to describe the sample.

### 2.3. ESG Instrumentation and Normalization Procedures

We downloaded each firm's 10-K filings to determine their true company name, and who regulated them, SEC, FINRA, NASD, or another organization. We used their actual name in combination with their subsidiaries and advertised names to collect the data. Artificial intelligence (AI) and internet searches were used to collect data to test the hypotheses. AI was used to search for legal proceedings of each firm in terms of documented regulatory violations as well as customer complaints decided against the firm. We searched the 10-K forms of each firm from the SEC to determine if specific legal proceedings were disclosed (https://www.sec.gov/edgar/search-and-access, accessed on 25 May 2024). Descriptive statistics were used to show what was reported versus what was found to be accurate.

We needed to collect ESG scores from reputable providers to answer our RQs and test our hypotheses; calculating ESG scores was beyond our RQ scope and available time. We leveraged the work of Dimmelmeier from 2023 [64] to select several reputable ESG rating companies. He conducted a meta-analysis of 128 ESG rating providers worldwide and evaluated the quality and usefulness of their indexes. His scope was global. We selected the top 10 ESG providers from his metadata.

Our analysis using Dimmelmeier's data is displayed in Figure 1. We transformed his data into positive and negative user rating scores. We looked for high scores in Figure 1, close to or above the +50 cutoff, shown in green and blue lines with shading. We did not specifically analyze the negative ratings since the overall results clustered the better-rated providers on the positive responses with corresponding lower negative trends. In other words, higher performers in quality were also higher in usefulness, with fewer low scores than their peers. Based on our results, we selected the following providers: CDP, MSCI, and Sustainalytics. CDP Climate, Water & Forests scored 54 in usefulness and 62 in quality. MSCI ESG was scored by participants at 40 for usefulness and 47 for quality. Sustainalytics' ESG Risk was the highest, being slightly beyond CDO on the usefulness factor, at 60, and comparable to the other two, at 53, for quality. Based on our analysis and transformation of Dimmelmeier's data, CDP, MSCI, and Sustainanalyics scored the highest in usefulness and quality.

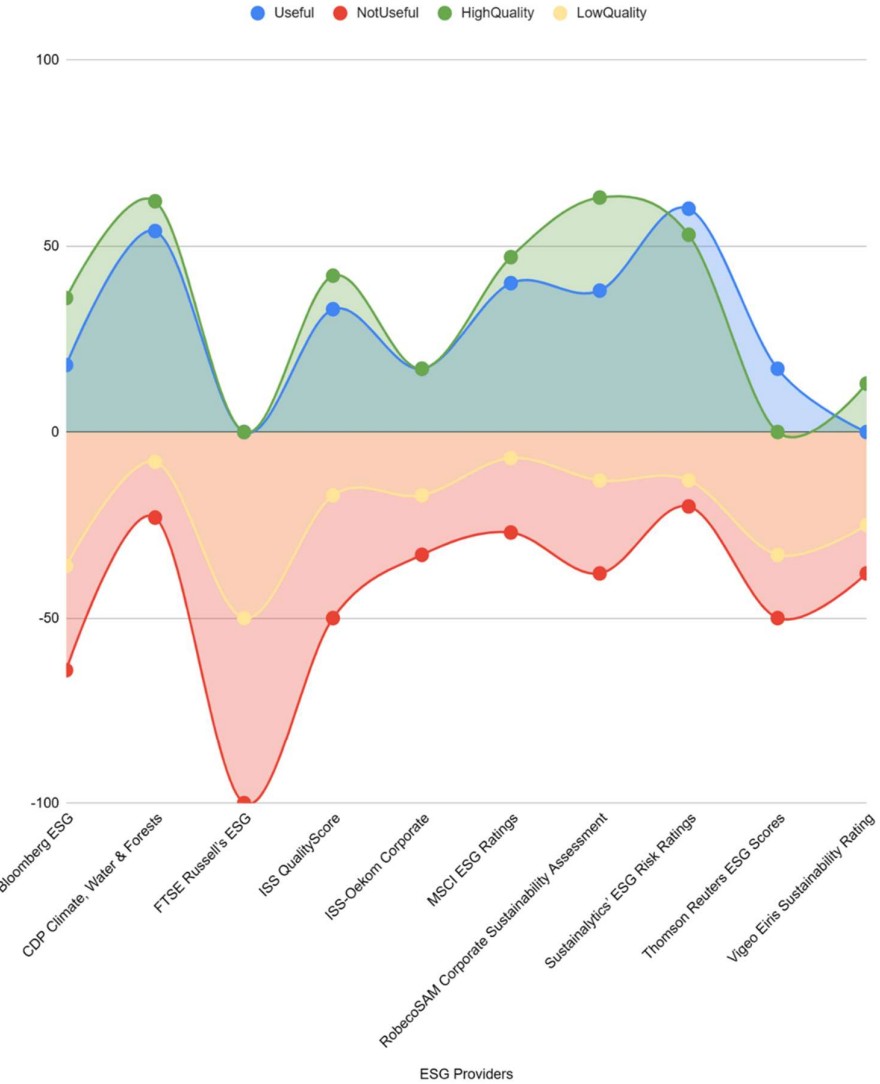

**Figure 1.** Descriptive comparison of ESG index companies in alphabetical order (N = 10).

We converted the Sustainability ESG risk scores into a 100 base index using the formula 1-ESG risk. We transformed the MSCI ESG scores, rated like bonds (AAA, AA, A, BBB, BB, B, C) into 1–7 intervals, and then normalized them to a 100 base. We normalized the Sustain Analytics ESG risk score to a positive ESG value. This was originally described as a negative risk, where higher was worse. We subtracted the ESG score from 100 to produce a normalized ESG rating comparable to those of the other providers.

Unfortunately, very few of our sampled firms had ESG ratings by CDP. None of the other ESG firms with high quality and usefulness scores had ratings for most of our sample. Therefore, we conducted auxiliary research to find an ESG provider substitute. We selected CSRhub as a proxy for CDP, because their scores were similar for other companies beyond our sample, and they explained their methodology in detail to increase their credibility (https://www.csrhub.com/csrhub-esg-ratings-methodology, accessed on 25 May 2024). CSRhub claimed they used 947 third-party rating sources to develop CSR ratings for 56,271 companies spanning 210 countries. They revealed that they cannot calculate reliable CSR ratings for 19,077 companies because they lack sufficient credible third-party rating data sources.

The ESG rating schema developed by CSRhub could be described as a meta-index or composite score derived from ratings published by other credible recommender systems, including Glassdoor, S&P Global, etc., which reduces bias due to the law of large sample size averages, since most of the sources come from 20 or more different rating companies,

each having multiple sources of their data, respectively. We developed the conceptual model in Figure 2 to guide our interpretation of their ESG ratings. None of the ESG providers provided detailed scores for every SDG, GRI, or ESG item level, so we could only compare them at the index level.

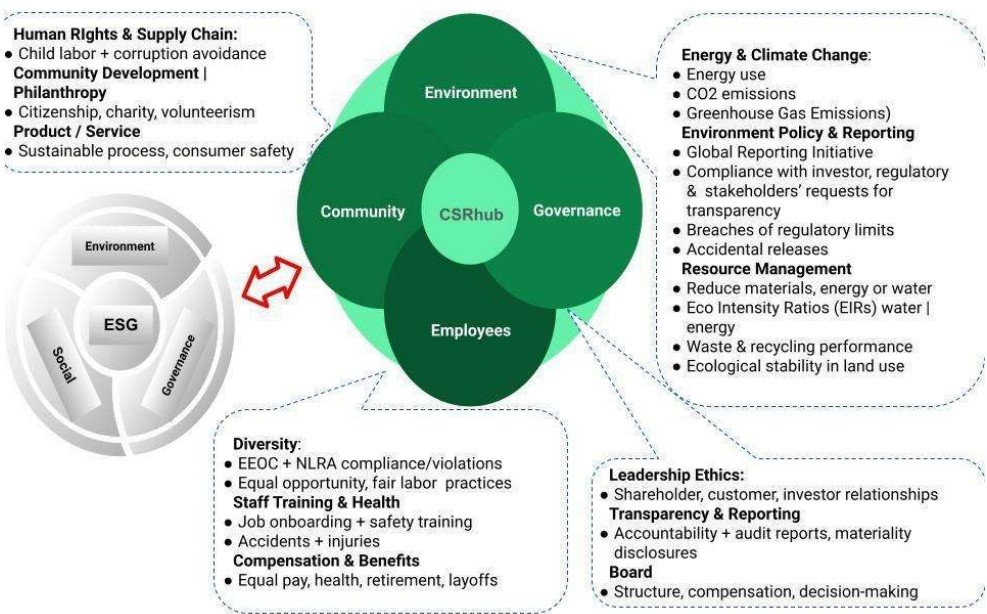

**Figure 2.** Authors' conceptual model of CSR four-factor 12-item ESG-like structure.

The CSRhub rating is not a precise ESG score, since it focuses on corporate social responsibility (CSR), of which ESG factors fit well into their schema. Their CSR typology has four factors: Environment, Community (social), Governance, and Employees, which roughly corresponds to ESG. Figure 2 presents our conceptual model of how the four-factor 12-item CSRhub typology corresponds to the traditional ESG structure. CSRhub's CSR environment factor measures these items: Environment policy and reporting; energy and climate change; resource management. Their community factor items are as follows: Human rights and supply chain; community development and philanthropy; product (referring to ability to create products/services using sustainable strategies like recycled containers and electronic paperwork delivery). The employees factor contains items that correspond to the social and governance categories in the traditional ESG structure. Employees captures measurements for these items: Hiring diversity and labor rights; staff training, safety, and health; compensation and benefits (refers to equal pay for same work, maternity/paternity leave, layoff policies, retirement plans). The CSR governance factor contains these items: Leadership ethics; transparency and reporting; board (composition and compensation strategy).

### 2.4. Data Collection Sources and Procedures

Our primary sources for researching misconduct and arbitration legal proceeding data were the U.S.-based government sanctioned regulatory agencies. The Financial Industry Regulatory Authority (FINRA) is one of three non-government self-regulating non-profit organizations overseeing the U.S. financial asset management industry, which corresponds to our targeted population. FINRA regulates approximately 627,637 registered representatives (RR) practicing as financial advisors and roughly 3435 firms [12]. The U.S. Securities and Exchange Commission (SEC) oversees firms with over $110 million AUM, including nearly 30,888 firms and 99,406 investment advisor representatives (IAR's) [13]. However, financial legal proceedings are commonly reported by FINRA. The North American Securities Administrators Association (NASAA) controls the remaining 17,685 firms along with unregistered FAs in the U.S., as well as in Canada and Mexico [14]. Various other U.S

government agencies assist in regulating the financial market, including the U.S. Federal Reserve, and the National Association of Securities Dealers (NASD). Some legal processes against financial asset management firms were recorded in the U.S. Courts (Federal, District, State, County). FINRA seems to be the aggregator of legal proceedings outside the U.S. Courts, but there is some overlap between FINRA and the Courts especially for appeals; therefore, we excluded the cases we found in the U.S. Courts as many were iterative appeals. We substantiated this choice because our RQs were focused on legal proceeding disclosures related to GRI anti-trust governance factor—appeals were beyond the scope of the present research.

We used AI to search for legal proceedings of any type associated with the legal name of the firms in our sample named as a defendant or respondent, including their subsidiaries, based on the U.S. SEC 10-K filing (which lists the legal name and affiliate companies). Our definition of legal proceedings was taken from GRI [17], namely:

"Number of legal actions pending or completed. . ." (GRI 206-1).

A misconduct legal proceeding outcome was defined as a consent, admittance, finding, or award against one of the firms where a fine, penalty, or sanction was levied by the relevant regulatory authority and then paid or acted on by the firm. Misconduct legal proceedings was the result of regulatory compliance investigations. An arbitration legal proceeding outcome was an award decided against the firm, which usually resulted in compensatory damages, fees, and sometimes punitive damages paid by the firm. Arbitrations resulted from firm clients complaining about unethical behavior, particularly failing to observe the customer's best interests or fiduciary misconduct. In other words, we did not count any legal proceeding unless it was final, decided against the firm, and formally reported by a recognized regulator. These data are indisputable and factual—anyone can verify them (see Appendix A).

To avoid duplicating the counts, we primarily extracted public data from FINRA or SEC, and we checked carefully using sorts based on date, case name, penalty, sanctions, and fine amount to ensure there were no duplicates in our results. The resulting evidence sources were cited, or the data were added to Appendix A.

### 3. Results

*3.1. Preliminary Results*

The final sample contained 14 firms, of which two companies were removed (The Capital Group and PGIM) due to a lack of consistent data across multiple sources. The AUM of the firms ranged from $1 trillion (Northern Trust Corp, Wellington Management, and TIAA-CREF) to $10.8 trillion (BlackRock Inc., New York, NY, USA). The descriptive statistics of the variables of interest are listed in Table 1. The ESG scores from the three independent providers are based on a 0 to 100 scale (keeping in mind that some were transformed, such as the MSCI bond-like ratings into normalized ratings).

**Table 1.** Descriptive characteristics of sample data (N = 14).

| Estimate | MSCI ESG | Sustain Analytics 1-ESG | CSRhub ESG | Misconducts | Arbitrations |
|---|---|---|---|---|---|
| Valid | 10 | 10 | 14 | 14 | 14 |
| Missing | 4 | 4 | 0 | 0 | 0 |
| Median | 78.6 | 76.3 | 92.0 | 27.0 | 6.0 |
| Mean | 75.7 | 77.3 | 85.4 | 328.4 | 131.1 |
| SD | 13.5 | 2.9 | 15.6 | 569.8 | 295.3 |
| Minimum | 42.9 | 72.9 | 44.0 | 6.0 | 1.0 |
| Maximum | 85.7 | 81.6 | 100.0 | 1758.0 | 986.0 |

The ESG scores ranged from 44 to 100 across all providers. There were four firms without ESG scores from MSCI and Sustain Analytics. Descriptively, the three ESG scoring data seem similar, i.e., the MSCI mean was 75.7 with a standard deviation (SD) of 13.5, Sustain was 77.3 (SD = 2.9), and CSRhub M = 85.4 (SD = 15.6). The higher median of 92 for CSRhub can be interpreted as indicating that most of the ESG ratings were higher in the distribution than the other two, with the MSCI median at 78.6, and Sustain at 76.3 (the latter were very similar).

The misconduct legal proceedings decided against firms had significant variation; the mean was 328.4 with a huge SD of 569.8, the minimum was 6 (Wellington Management) and the maximum was 1758 (Morgan Stanley). The large SD of 569.8 and low median of 27 indicated that a few companies in the data impacted the dispersion—this is a reg flag for statisticians and auditors. The numbers of arbitration legal proceedings decided against firms were also high, with a mean of 131.1 (SD = 295.3), ranging from a low of 1 case (J.P. Morgan Chase & Co) to 986 (Morgan Stanley). The median of 6 along with the high SD of 295.3 again suggests a few firms skewed these results.

### 3.2. How Consistent Are ESG Ratings from Different Providers for the Same Firms

The descriptive statistics did not reveal the reliability or consistency of ESG scores from the providers for the same firms. Bivariate correlation tests were used to answer H1 (Popular ESG ratings do not consistently measure anti-corruption governance because they differ significantly for the same financial asset management firms). The results are summarized in Table 2 with significant correlations flagged with asterisks. The misconduct and arbitration legal proceedings counts were also reported in this table but are discussed later.

Only two ESG providers had consistent similar scores, MSCI and CSRhub, with a significant positive correlation of r = +0.891 ($p < 0.001$; VS-MPR = 165.92). We can interpret this as follows: 89% of the ESG scores from MSCI and CSRhub will be approximately the same 99 times out of 100 if we were to sample more firms from the same financial industry population. We can also observe from Table 2 that the lower and upper 99% *p*-value confidence interval of (0.05, 0.99) does not contain zero, indicating this was an acceptable result. The high Vovk-Selke maximum p-ratio (VS-MPR) indicates that this test is 166 times more likely to result in a significant positive correlation between MSCI and CSRhub ESG scores. According to Selke et al. [20], the VS-MPR is robust; it is calculated using a formula based on the reciprocal of the log for the *p*-value to approximate the distribution shape under the null versus alternate hypothesis conditions.

**Table 2.** Bivariate correlation between ESG scores, misconducts, and arbitrations.

| Bivariate | Comparisons | n | Coefficient | *p* | Lower 99% CI | Upper 99% CI | VS-MPR |
|---|---|---|---|---|---|---|---|
| MSCI ESG | Sustain Analytics 1-ESG | 9 | −0.069 | 0.57 | −0.926 | 0.467 | 1 |
| MSCI ESG | CSRhub ESG | 10 | 0.891 *** | $2.698 \times 10^{-4}$ | 0.05 | 0.99 | 165.92 |
| MSCI ESG | Arbitration | 10 | 0.266 | 0.229 | −0.915 | 0.756 | 1.09 |
| MSCI ESG | Misconduct | 10 | 0.323 | 0.181 | −0.415 | 0.771 | 1.188 |
| Sustain Analytics 1-ESG | CSRhub ESG | 10 | 0.105 | 0.386 | −0.725 | 0.537 | 1 |
| Sustain Analytics 1-ESG | Arbitration | 10 | −0.555 | 0.952 | −0.866 | −0.339 | 1 |
| Sustain Analytics 1-ESG | Misconduct | 10 | −0.59 | 0.964 | −0.874 | −0.037 | 1 |
| CSRhub ESG | Arbitration | 14 | 0.053 | 0.428 | −0.873 | 0.276 | 1 |
| CSRhub ESG | Misconduct | 14 | 0.203 | 0.243 | −0.223 | 0.447 | 1.07 |
| Arbitration | Misconduct | 14 | 0.897 *** | $7.028 \times 10^{-6}$ | −0.195 | 0.995 | 4411.396 |

Note: Significant correlations marked as *** $p < 0.001$.

We can also observe from Table 2 that the MSCI ESG score was not positively correlated with the Sustain Analytics ESG score for the same firms. Similarly, the CSRhub ESG the score was not positively correlated with the Sustain Analytics ESG score for the same firms. Based on these results, we can accept H1 but note that the MSCI and CSRhub ESG scores were positively correlated for the same firms. Still, we must reject H1 for the other provider, Sustain Analytics. We must also bear in mind that the MSCI ESG scores were transformed from bond-like ratings and rescaled—if MSCI had created a 0–100 ESG score, perhaps the correlation estimates would have differed.

We can also observe a significant positive correlation of r = +0.897 between misconduct and arbitrations in Table 2, with $p < 0.001$ and VS-MPR = 4411.4 (rounded). Based on the VS-MPR, we could conclude that this result is 4411 times more likely to reveal that misconduct and arbitration legal proceedings are positively correlated, i.e., 99 times out of 100, if we re-sampled all firms in this population. We will build on this notion later. At this point there is no doubt that popular ESG scores significantly differed across providers.

### 3.3. Misconduct and Arbitration Adversarial Legal Process Outcomes Reflected in ESG Scores

We leveraged Bayesian correlation and descriptive statistics to answer H2 (number of adversarial legal outcomes—misconduct, unethical behavior, regulatory violations—of large financial asset management firms accurately reflected in ESG scores). Table 3 summarizes the data used to test H2, and we will draw upon some correlations reported earlier in Table 2.

We can observe from Table 3 that the misconduct and arbitration counts are unusually high for firms in the middle of the data. The counts of cases were 300–1000 times higher for some firms (1758 versus 6 misconducts = 298 times higher, and 968 versus 1 arbitration = 968 times higher). Another record from Table 3 had similar unusual dispersion (1322 versus 6 misconducts = 220 times higher, 614 versus 1 arbitration = 614 times higher). These are red flags, i.e., a problem has been detected. Due to these anomalies (red flags), we leveraged Bayesian correlation and descriptive statistics to answer H2 (number of adversarial legal outcomes—misconduct, unethical behavior, regulatory violations—of large financial asset management firms were accurately reflected in popular ESG scores).

Hypothesis H2 was supported. The ESG scores did not reflect legal process misconduct or arbitration outcomes. If they did, we would have seen a few minimal correlations between the ESG scores and either misconduct or arbitration legal case counts. The results of the Bayesian correlation are summarized in Table 4.

We can start the analysis of Table 4 by noting the corroboration with earlier parametric correlation results, as shown in Table 2. The MSCI and CSRhub ESG scores had a +0.891 correlation (BF10 = 131.1), with 99% confidence intervals (0.456, 0.972). We do not need to calculate the VS-MPR, because the Bayesian correlation is robust. The BF10 odds of the *p*-value being true were lower than previous correlation because the Bayesian and bootstrapping increased the sample size. Almost every statistical coefficient in some way incorporates standard deviation, which uses sample size in the denominator of the algorithms. By increasing the sample size, we increase the power and precision when all other parameters remain unchanged.

We can also observe from Table 4 that legal process misconduct and arbitration counts were positively correlated at +0.897, identical to the results reported in Table 2. Still, the Bayesian $BF_{+0}$ was 3005 with 99% confidence intervals (0.617, 0.965). In the earlier results, we observed the same interpretation, i.e., a significant positive correlation of +0.897, with a VS-MPR of 4411 odds in favor of the alternate hypothesis. Here, we have a comparable odds ratio of 3005, meaning that the likelihood that misconduct and arbitration legal process counts will be positively related is 3005 if data were collected again from this population (in 99 out of every 100 samples).

**Table 3.** Financial asset management company data summary (N = 14).

| Company | Approximate AUM | Misconduct | Arbitration | CSRhub ESG | Sustain Analytics 1-ESG | MSCI ESG |
|---|---|---|---|---|---|---|
| BlackRock Inc., New York, NY, USA | $10,800,000,000,000 | 7 | 1 | 94 | 81.6 | 71.4 |
| Vanguard Group, New York, NY, USA | $8,600,000,000,000 | 13 | 18 | 61 | | |
| Fidelity National Financial, Inc., New York, NY, USA | $5,300,000,000,000 | 20 | 154 | 44 | 75.6 | 42.9 |
| State Street Corporation, New York, NY, USA | $4,340,000,000,000 | 34 | 15 | 95 | 76.8 | 85.7 |
| Morgan Stanley, New York, NY, USA | $4,100,000,000,000 | 1758 | 986 | 91 | 75.2 | 85.7 |
| UBS Group AG, New York, NY, USA | $4,020,000,000,000 | 1322 | 614 | 92 | 72.9 | 85.7 |
| J.P. Morgan Chase & Co, New York, NY, USA | $3,600,000,000,000 | 739 | 1 | 92 | | 71.4 |
| The Goldman Sachs Group, Inc., New York, NY, USA | $2,850,000,000,000 | 584 | 24 | 86 | 75.8 | 71.4 |
| The Bank of New York Mellon Corp., New York, NY, USA | $1,800,000,000,000 | 41 | 3 | 93 | 81 | 85.7 |
| Invesco Ltd., New York, NY, USA | $1,410,000,000,000 | 17 | 2 | 80 | 78.1 | |
| Franklin Resources, Inc., New York, NY, USA | $1,390,000,000,000 | 37 | 2 | 77 | 80.5 | 71.4 |
| Northern Trust Corp., New York, NY, USA | $1,000,000,000,000 | 7 | 4 | 92 | 75.1 | 85.7 |
| Wellington Management, New York, NY, USA | $1,000,000,000,000 | 6 | 7 | 98 | | |
| TIAA-CREF, New York, NY, USA | $1,000,000,000,000 | 12 | 5 | 100 | | |

Sources: Firm name and AUM retrieved from 10-K filing for 2023 data (SEC, 2024); FINRA and SEC data (see Appendix A); CSRhub ESG data retrieved 24 May 2024, from https://www.csrhub.com/, MSCI ESG data retrieved 24 May 2024, from: https://www.msci.com/; Sustainanalytics data retrieved 24 May 2024, from: https://www.sustainalytics.com/esg-rating/.

**Table 4.** Bayesian correlation of ESG, misconducts, and arbitrations (N = 14,000 ****).

| Bivariate | Comparison | Coefficient | BF$_{+0}$ | Lower 99% CI | Upper 99% CI |
|---|---|---|---|---|---|
| MSCI ESG | Sustain Analytics 1-ESG | −0.069 | 0.359 | 0.009 | 0.615 |
| MSCI ESG | CSRhub ESG | 0.891 *** | 131.101 | 0.456 | 0.972 |
| MSCI ESG | Misconduct | 0.323 | 0.892 | 0.021 | 0.742 |
| MSCI ESG | Arbitration | 0.266 | 0.743 | 0.018 | 0.719 |
| Sustain Analytics 1-ESG | CSRhub ESG | 0.105 | 0.485 | 0.012 | 0.656 |
| Sustain Analytics 1-ESG | Misconduct | −0.59 | 0.165 | 0.004 | 0.382 |
| Sustain Analytics 1-ESG | Arbitration | −0.555 | 0.171 | 0.004 | 0.397 |
| CSRhub ESG | Misconduct | 0.203 | 0.61 | 0.015 | 0.635 |
| CSRhub ESG | Arbitration | 0.053 | 0.379 | 0.009 | 0.565 |
| Misconduct | Arbitration | 0.897 *** | 3005.193 | 0.617 | 0.965 |

Note: Significant correlations marked as *** $p < 0.001$; **** bootstrapped 1000.

This may be interpreted as showing that Bayesian correlation will not change basic correlation coefficients, but that the reliability, validity, and credibility will increase due to the sample size and robust algorithms. The sample size of 14000 reflects that bootstrapping was applied to replicate the sampled data values.

We can see that the BF10 of 131 is statistically close to the earlier VS-MPR of 166. This can be interpreted similarly: the positive correlation of MSCI and CSRhub ESG scores is 133 times more likely in 99% of 100 samples taken from the same population of financial asset management firms. We can also see that the corroboration that the Sustain Analytics ESG score was unrelated to the other providers. None of the ESG scores was related to the misconduct or arbitration counts. All BF10 coefficients were less than 1, which can be interpreted as indicating that there was no relationship or association between them. The ESG ratings from providers did not reflect misconduct or arbitration cases decided against the firms. Consequently, there was very strong evidence to accept H2.

*3.4. Data Content Analytics*

Here, we further analyze the evidence to understand why the hypotheses were supported by the data. We observed the red flags in Tables 2 and 4 where anticipated correlations were not found, and legal proceedings were astronomically high for a few firms compared to others in the sample. We can share that there were no significant nonparametric correlations between AUM or number of employees and the ESG ratings (results not reported).

To further investigate these anomalies, we searched the SEC for 2023 year-end versions of the 10-K form filings (or similar such as 20-K), for all firms in the sample. Only a few firms disclosed misconduct and arbitrations in sweeping statements acknowledging that there had been legal proceedings. Here is the legal proceedings disclosure by Blackrock for year-end 2023 (https://www.sec.gov/ix?doc=/Archives/edgar/data/1364742/000095017024019271/blk-20231231.htm, accessed on 24 May 2024): "Legal Proceedings. From time to time, BlackRock receives subpoenas or other requests for information from various US federal and state governmental and regulatory authorities and international governmental and regulatory authorities in connection with industry-wide or other investigations or proceedings. It is BlackRock's policy to cooperate fully with such matters. BlackRock has been responding to requests from the SEC concerning a publicly reported, industry-wide investigation of investment advisers' compliance with record retention requirements for certain electronic communications. BlackRock is cooperating with the SEC's investigation. Certain subsidiaries and employees have been named defendants in various legal actions, including arbitrations and other litigation arising in connection with BlackRock's activities. Additionally, BlackRock-advised investment portfolios may be subject to lawsuits, which could potentially harm the applicable portfolio's investment returns or result in the Company being liable to the portfolios for any resulting damages."

Ironically, Blackrock (the largest asset management firm in the world) had the best outcome in terms of the fewest legal misconduct and customer arbitration legal cases decided against them. We originally expected the opposite for Blackrock based on the literature review. Blackrock had only seven misconduct disclosures with fines paid and one arbitration legal proceeding decided against them.

Other samples in our analysis were much worse. We investigated the top two high legal proceeding count firms, UBS and Morgan Stanley, by checking their similar filings, to determine if the misconduct and arbitration results had been cited. UBS had approximately twice as misconduct regulatory fines and customer arbitration cases decided against them compared to the next runner up (J.P. Morgan Chase). UBS AG (a Switzerland-based company operating in the USA) had many pages of explanations concerning legal proceedings, with extensive explanations of how they were addressing their problems, but no specific mention of the number of misconduct or arbitration cases decided against them. Here is one typical quote from their most recent 20-K (https://www.sec.gov/ix?doc=/Archives/edgar/data/1364742/000095017024019271/blk-20231231.htm, accessed on 24 May 2024):

"As a global financial services firm operating in more than 50 countries, we are subject to many different legal, tax and regulatory regimes, including extensive regulatory oversight, and are exposed to significant liability risk. We are subject to many claims, disputes, legal proceedings and government investigations, and we expect that our ongoing business activities will continue to give rise to such matters in the future. . .. Since the financial crisis of 2008, we have been subject to significant regulatory requirements, including recovery and resolution planning, changes in capital and prudential standards, changes in taxation regimes because of changes in governmental administrations, new and revised market standards and fiduciary duties, as well as new and developing environmental, social and governance (ESG) standards and requirements."

Morgan Stanley was the worst performer in the sample, with significantly higher misconduct regulatory fines and customer arbitration cases decided against them. Their misconduct and arbitration legal proceeding counts were approximately three times larger than the next runner up (J.P. Morgan Chase). If we compare the misconduct regulatory fines and customer arbitration decisions against Morgan Stanley to the best performer in our sample, Blackrock, we observed that Morgan Stanley's misconducts were 251 times higher, and their customer arbitrations decided against them were 986 times higher than those of Blackrock.

We discovered that Morgan Stanley made generic statements in their most recent 10-K filing, but they did not mention misconduct or arbitration proceedings decided against them. Here is a relevant quote from their most recent filing for year-end 2023 (https://www.morganstanley.com/content/dam/msdotcom/en/about-us-ir/shareholder/10q0324.pdf, accessed on 24 May 2024): "In addition to the matters described below, in the ordinary course of business, the Firm has been named, from time to time, as a defendant in various legal actions, including arbitrations, class actions, and other litigation, arising in connection with its activities as a globally diversified financial services institution. Certain actual or threatened legal actions include claims for substantial compensatory and punitive damages or claims for indeterminate damages. In some cases, the third-party entities that are, or would otherwise be, the primary defendants in such cases are bankrupt, in financial distress, or may not honor applicable indemnification obligations. These actions have included, but are not limited to, antitrust claims, claims under various False Claims Act statutes, and matters arising from our sales and trading businesses and our activities in the capital markets".

However, Morgan Stanley did not mention ESG compliance in their 2023 year-end filing, as UBS had done—instead, they referred to a 2022 report about their ESG compliance.

In summary, we found that the ESG providers likely did not find the legal proceedings to be evidence of misconduct and arbitrations decided against the firms they had rated favorably. This illustrated two problems in this practice. First, ESG providers cannot report what they do not know—which may lead to unreliable or inaccurate products. Second, the regulators in the financial asset management industry do not mandate or enforce ESG rating firms to evaluate legal proceedings such as misconduct and arbitration rulings against firms. The evidence speaks for itself, and it is credible.

*3.5. Root Cause Analysis*

We conducted additional ad hoc analyses to uncover the root cause of why the ESG providers did not meet the SDG, ESG and GRI 206-1 standard. We produced a line chart to visually contrast the ESG scores, misconduct and arbitration counts from the data in Table 3, as illustrated in Figure 3. The sampled firms were placed on the $x$ axis, ordered by lowest number of combined misconduct and arbitration proceedings to highest, left to right. The left vertical $y$ axis was set to the number of misconducts and arbitration legal proceedings on a 0 to 2000 ratio scale. The ESG scores were plotted against a third $z$ axis, shown on the right as the ESG score from 0 to 100. Some line series are not connected due to missing ESG score data.

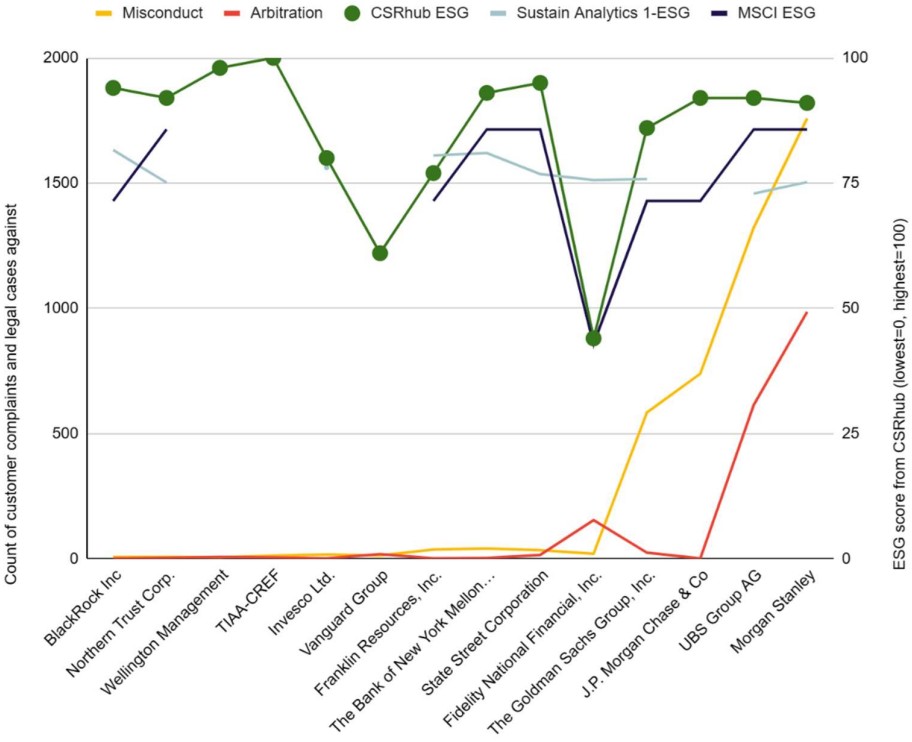

**Figure 3.** Asset management company ESG ratings versus arbitration and misconduct legal proceedings (N = 14).

The line chart in Figure 3 corroborates our interpretations of Tables 2–4, i.e., that ESG scores are not correlated with legal proceedings. We can visually observe relationships between MSCI and CSRhub ESG scores and between misconduct and arbitration legal proceedings. The problem detected by this analysis was the ESG scores do not reflect documented legal proceedings against firms, such as misconduct or arbitrations. Figure 3 can be interpreted as showing that firms on the left have fewer legal proceedings while firms on the right have the most.

We can observe two interesting results from Figure 3. The line series representing CSRhub (green line with circles on data values) is parallel with MSCI ESG score (the blue line, noting the open points where data were missing). We can see there is some similarity between Sustain Analytics (light blue line) and these two series (MSCI and CSRhub). However, it is clear there is a disconnect between ESG scores and misconduct or arbitration legal process outcomes. The disconnect is obvious, because the misconducts (yellow line) and arbitration legal proceedings (red line) ought to be inversely related to the other series. The yellow and red line series for misconduct and arbitrations are very high for UBS and Morgan Stanely. The second disconnect is that firms with low numbers of misconduct and arbitration legal proceedings were rated lower by CSRhub and MSCI, namely Vanguard and Fidelity.

## 4. Discussion and Conclusions

In this study, the authors focused on investigating whether the popular ESG scores were reliable and consistent in rating anti-corruption factors, which are part of the ESG sustainability framework based on the GRI 206-1 standard [17]. Based on our sample, ESG providers did not measure anti-corruption governance, and their ratings were not consistent for the same firms, at least in the financial asset management industry. We can conclude that this study was successful, in as far as the RQs were answered:

- RQ1: Do common ESG sustainability indexes of large financial asset management firms with AUM > $1 trillion accurately measure anti-corruption governance? Answer = no (indisputable evidence for H1);
- RQ2: Are large financial asset management firms disclosing anti-corruption governance to comply with GRI 206-1 (number and outcomes of legal actions)? Answer = no (based on rigorous Bayesian correlation H2 testing).

We came to the above conclusions by evaluating a sample of 14 financial asset management firms with an AUM of at least $1 trillion drawn from the USA. The goal was to determine if the ESG rating providers were collecting the relevant data to inform shareholders regarding the degree to which financial asset management firms complied with the 17 United Nations sustainability development goals (SDGs), particularly the governance factor in ESG, as measured by anti-corruption using the GRI 206-1 standard. According to the evidence, all ESG providers and firms failed to meet the above standards. We went much deeper to investigate the why of these findings, as discussed below.

### 4.1. Ad Hoc Analysis and Comparison of Findings to Extant Literature

As an ad hoc test, we filtered out the two firms with extremely high numbers of misconduct and arbitration legal proceedings (UBS and Morgan Stanley) from the rigorous Bayesian correlation analysis (see Table 4) and Figure 2 (chart). Two interestingly different results emerged. First, there were no significant correlations between misconduct and arbitration legal proceedings for any of the remaining 12 firms in the sample. Second, The MSCI ESG rating was significantly negatively correlated with arbitration legal proceedings at $-0.89$ ($p = 0.009$, VS-MPR = 8.771). The high VS-MPR odds can be interpreted as this negative correlation being almost nine times more likely to be seen than a positive or no correlation in 99 out of 100 samples from the population.

We may interpret this as showing that the MSCI ESG scores are still inaccurate in measuring governance anti-corruption compliance. However, removing the two firms with the largest numbers of misconduct and arbitration legal proceedings substantially changed the distribution, because customer complaints (arbitration legal proceedings) are not linked to regulatory fines (misconduct). This suggests conditional logic that only some of the 12 firms have correlations between customer dissatisfaction (arbitrations) and industry regulatory fines (misconduct). The top two offenders, UBS and Morgan Stanley, have a consistent correlation between customer complaints decided against them (arbitrations) and regulatory fines (misconduct), while other firms have a negligent correlation.

We found a viable explanation in the literature for this ad hoc insight. Independent researchers and attorneys examined one firm in our sample (Morgan Stanely) to investigate gaming the FINRA system and deleting misconduct records. The Public Investors Advocate Bar Association Foundation (PIABA) asserted that deleting financial misconduct records (called expungements) should be stopped, because financial firms were gaming FINRA's arbitration system [65]. Doss and Bragança explained that these unethical firms include a nominal $1 demand in damages (but later withdraw that claim) to reduce the number of arbitrators reviewing expungement requests and to make it cheaper for firms to get customer complaints deleted [65]. This could explain why misconduct counts by regulatory authorities were higher than arbitration legal proceedings from customers decided against the firms. Doss and Bragança observed that "Expungement requests are being granted based upon one-sided and possibly false evidence presented to arbitrators" (pp. 4–5).

Egan, Mavos, and Seru analyzed 8 million records from FINRA, finding that 70% of financial advisors had misconduct records, a third of them were repeat offenders, and prior offenders were five times more likely to be repeat offenders [66]. They observed that certain firms have many more misconduct records relative to their size. UBS (fourth spot with 19%) and Morgan Stanley (tenth spot with 15.8%) were listed by Egan, Mavos, and Seru [66] as being in the top 10 firms disciplined for misconduct. The firm they found with the most misconduct was Oppenheimer and Co. No other firms in our sample were listed by Egan, Mavos, and Seru [66] for high relative misconduct. Those authors concluded that "some

firms specialize in misconduct (p. 34), and they continue to employ and hire financial advisors who have misconduct records with nowhere else to work."

Honigsberg and Jacob [67] extended the work of Egan et al. by analyzing 6660 unique expungement requests. They found "successful expungements predict future misconduct; brokers with prior expungements are 3.3 times as likely to engage in new misconduct as the average broker" (p. 4). They also stated that Morgan Stanley had the "highest number of expungements, 572, relative to misconducts" of all firms (p. 41). Honigsberg and Jacob argued that financial asset management firms were abusing the expungement process by deleting misconduct records, which perpetuates unethical behavior by rewarding it.

A reasonable decision-maker would not likely ignore such unethical behavior. However, ESG providers are ignoring unethical behavior. The best example goes back to Volkswagen's actions, discussed earlier, which Siano, Vollero, Conte, and Amabile stated were not justifiable under any ethical principle, including the ESG sustainability framework. Another dimension would be the sheer volume of misconduct and arbitration decisions against financial asset management firms. Volkswagen had one legal proceeding decided against them by regulators, resulting in a dramatic income decline of 1.7 billion euros [14]. We assert that if such a negative impact were imposed against firms in our sample who failed to disclose fines paid and legal misconduct cases decided against them, shareholders would quickly sell their investments due to a lack of trust and fear of investment income loss. Consider, for example, how shareholders would react if they knew UBS had 1322 misconduct disclosures and 614 arbitration decisions against them, or Morgan Stanley with 1758 misconduct records and 986 customer arbitration decisions decided against them.

The seriousness of the arbitration and misconduct unethical behavior legal proceeding records must also be considered in weighing ESG and regulatory responsibilities. Most, if not all, of the misconduct legal proceedings reviewed in this study, amounted to clear unethical behavior. For example, FINRA recently fined Morgan Stanley $1.6 million for the firm's repeated failures to close out failed inter-dealer municipal security transactions promptly, and FINRA stated they had previously sanctioned Morgan Stanley for supervisory failure. Miami, FL-based Blum Law Group [68,69] has published numerous studies about misconduct—they stated that FINRA fined Morgan Stanley "$5 million for failing to accurately supervise many of its brokers. . ." (para. 2). Blum Law Group [69] complained that "a powerhouse brokerage firm that has been around for more than 75 years and has more than $1.65 trillion in client assets that it manages, they should have known better" (para. 2).

Based on the above, we may conclude that ESG providers did not read the misconduct or arbitration data. We could not find evidence that any ESG provider discovered relevant legal proceedings. It is illogical for ESG providers not to consider those misconduct and arbitration legal proceedings. Furthermore, we found no additional literature beyond the PIABA, Egan et al., and Honigsberg et al. studies, where researchers had found and analyzed financial misconduct or arbitration legal proceedings. Therefore, we suggest ESG providers and researchers use AI to help them locate the relevant data. We also recommend that regulators, particularly the SEC, oversee ESG providers to ensure that the requisite misconduct and arbitration data are being evaluated and that financial asset management firms accurately disclose their legal proceedings without using broad, sweeping statements.

### 4.2. Stakeholder Theory and Public Theory Point of View

There is usually another point of view on any findings, and we discuss two opposing theories here.

Stakeholder theory suggests that firms exist to serve their shareholders, which means maximizing profits that will be distributed to shareholders as dividends [26]. Under the stakeholder theory perspective, firms may be thought of as acting within their mission to avoid disclosing misconduct and arbitration decisions decided against them. Certainly, one could imagine if UBS and Morgan Stanley listed all of their misconduct and arbitration legal

proceedings decided against them in their prospectus, annual report, and other marketing brochures, the result would not be favorable for their shareholders.

Yet another perspective to stakeholder theory in this study is that some firms may have a well-thought-out strategy to diminish the seriousness of misconduct and arbitration legal proceedings decided against them. Suppose a firm has more than sufficient profit to pay for arbitration and misconduct fines, penalties, sanctions, and legal costs. In that case, this may be considered as a factor of production, a necessary operating expense.

We noted that Park and Jang [70] adopted this perspective as they explored how companies could develop strategies to comply with ESG. We are not trying to justify this behavior, but rather, we are trying to rationalize why this unethical behavior continues. It was clear from the data collection that the misconduct and arbitration legal proceeding decisions dated back decades. This is not a post-COVID phenomenon or a new regulatory initiative. It appears nondisclosure is part of the production formula, a standard operating procedure for some firms in the financial asset management population we sampled. It is certainly an interesting topic to research in the future.

A contrary perspective to stakeholder theory is based on the public interest theory. Public interest theory was found by Taylor and Whitton [71] to be ambiguous and pluralistic. In general, Taylor and Whitton [71] determined that the best explanation for the theory of public interest was to meet the common objectives valued by the community, create a fair and proper balance between individual and group interests, and ensure the delivery of democratic expectations. If we apply public interest theory to the findings of this study, we must reject our stakeholder theory proposition discussed above. Our argument in favor of public interest theory is if only the firm and its shareholders benefit from unethical disclosures for ESG ratings, that will be unfair to other firms who are honest, unfair to group investors such as pension plan managers, and unfair to individual retail investors who would not be warmed about the higher risk associated with firms who do not disclose fines and legal cases decided against them in their ESG scores. Whether the affected parties are non-shareholders is not the point—the lack of disclosure and transparency are contrary to public interest theory.

From the public interest theory standpoint, we could also assert that disclosing fines paid and legal cases decided against a firm would be relevant to report for an ESG score due to social coexistence and social norms in our understanding of global culture. While above we argued in favor of stakeholder theory, i.e., that a firm can consider misconduct fines, penalties, sanctions, and legal cost as a proprietary factor of production that need not be disclosed, we could contrast to that the need for society in general to absorb the losses for those negatively impacted by such nondisclosures. We could go so far as to juxtapose public interest theory as a tax to be imposed on firms who leverage non-transparency and do not disclose legal fines or cases against them. We mean that such firms, which other researchers have labeled unethical, ought to be taxed for this exception, just as a company or country that pollutes the environment must either pay a fine, tax, or tariff in the form of a carbon tax on exports or sales. Subsequently, in an accounting sense, such legal nondisclosures and potentially low ESG scores could be considered nontax deductible in the balance sheet, and it ought to be presented to shareholders and stakeholders as a vulnerability of the company.

Given that stakeholder theory states that firms should act in the best interests of shareholders, it is difficult to understand why a lay person, a shareholder, or retail investor, would not benefit from knowing about the frequency of misconduct and arbitration legal proceedings decided against the firm.

In closing, we remind readers that we do not see our role as recommending how firms may comply with regulations or how to satisfy their shareholders to support stakeholder theory. The scope of our research was limited to ESG compliance testing and, more specifically, the GRI 206-1 anti-corruption governance factor compliance, which was only one of the items in the governance factor of the GRI construct. We also caution readers that we had a small sample size of 14 firms. Despite applying robust Bayesian correlation with

a 1000 bootstrap procedure, we had a few missing ESG rating values, and we only sampled large firms with AUM of at least $1 trillion.

### 4.3. Recommendations for Overcoming ESG Rating Deficiencies

Our first recommendation is for the regulators of ESG producers and financial asset management firms. Broad sweeping statements in 10-K forms or marketing brochures do not accurately convey the facts of 1758 (Morgan Stanley) and 1322 (UBS) misconduct regulatory fine legal proceedings, or 986 (Morgan Stanley) and 614 (UBS) customer arbitration legal proceeding legal proceedings decided against the firms. It would not be reasonable for a shareholder, retail investor, or potential downstream supply chain provider not to wish to know the misconduct and arbitration legal proceedings decided against a firm, whether they are in the financial services sector or some other industry. This was a novel AI-assisted data collection approach because we could not find any studies beyond three papers where researchers revealed misconduct and arbitration data.

We could also suggest, in support of public interest theory [71], that retail investors, group investors, and consumers ought to be more aware of governance factors within the ESG ratings, including fines paid by firms and legal misconduct cases decided against firms. We might draw an imaginary parallel between the financial industry and marijuana, tobacco, or alcohol manufacturing companies. In the USA, the U.S. Surgeon General requires those companies to place consumer warnings on their products about known disclosures and potential side effects. From the public interest theory standpoint, we might suggest the U.S. Attorney General mandate financial asset management firms to place a label on their services disclosing the fines paid and misconduct legal process cases decided against them (perhaps at least the number of fines and cases with a URL to the details should a shareholder or retail investor which to look further before purchasing).

We also have recommendations for regulation of the ESG providers. Perhaps the providers could agree that they need selection criteria to determine what mandates ethical behavior, and only firms who meet that bar could be rated, with nothing said about the unqualified firms (to be fair to them based on stakeholder theory). That approach would also support public interest theory. The important question is how many instances of misconduct or arbitrations warrant being unqualified in scoring by an ESG provider or citing SDG ESG sustainability affiliations in marketing materials. Looking back at stakeholder theory, what would be reasonable for a stakeholder or retail investor to use as a criterion when considering purchasing the financial asset management firm's services? Would a layperson or wealthy investor be willing to overlook three or six or more misconduct and arbitration legal proceedings decided against the firm? Would the recency of those legal proceedings or fine amounts be part of that criterion?

The mathematical problem is that due to the large number of items being measured by GRI for each ESG factor, each item would only receive a 1–3 point relative weighting overall. That would mean that if a firm had more than three misconducts or more three arbitration legal proceedings combined, additional unethical behavior would not further decrease their score. Thus, the ethical behavior cutoff for ESG rating and SDG sustainability affiliation claims might be set at three combined legal proceedings. That is a low amount. None of the firms in our sample would meet that criterion, so none would qualify for an ESG rating or be in a position to state that they are aligned with SDG sustainability in their marketing promotions. Some ESG indexes are relative to the sample, using ratings against their peer, meaning that a poor performer could be rated well if other firms had lower performance. This might mean several unethical firms would receive scores implying they were ethical.

We also recommend that future researchers investigate the recency and number of misconduct and arbitration legal proceedings decided against firms to determine the extent of the unethical behavior and its impact on other factors. We will not try to address that dilemma here. Instead, we call for other researchers to pursue this worthy topic. We have different specific recommendations for researchers to consider in future studies. First, other

researchers should expand the sample size to evaluate more firms, not just those with AUM > \$1 trillion. It also makes sense to go beyond the USA and assess firms in other global regions.

Second, other researchers should develop better ESG measurement instruments in two categories. One category should be a simple instrument that small, medium, and large firms may use to ensure everyone is on the same playing field. This ESG measurement instrument should focus only on the essential ESG factors that strategically support sustainability, particularly human rights, honesty, ethics, and full disclosure. Hiding data cannot help anyone correctly measure ESG sustainability. The second category of ESG instrument that is needed is the GRI model, because it measures the relevant factors to meet sustainability. The GRI model ensures that ethical behavior and governance evidence are counted, which will prevent firms from promoting alignment with ESG without making any credible contribution to sustainability development goals.

Third, we recommend that additional methods be used. In our study, we used statistical method triangulation, three different correlation techniques, bootstrapping, data field counts using software, and qualitative root cause analysis of the textual records. We also attempted to corroborate or refute our findings with the extant literature. We strongly encourage researchers to introduce a dependent variable, such as shareholder satisfaction or profitability, to a quantitative analysis model, which would support robust multivariate analysis. If we ever reach a point when the ESG scores are considered valid, reliable, and credible, that could become one of the dependent variables for researchers to test. We discourage using surveys, because they are self-reported opinions with inherent bias and no guarantee the targeted population was sampled. Instead, we recommend evidence-based data collection.

Fourth, when considering our post hoc root cause analysis from the stakeholder theory and public interest theory perspectives, based on interpreting the rigorous Bayesian statistical results against the GRI 206-1 anti-corruption governance factor of the ESG construct, it was clear that one firm was a good performer, and two firms were significantly bad performers. Blackrock was the best performer in the sample. Morgan Stanley was the worst performer, with 1758 misconduct regulatory fines, 258 times higher than the best performer, Blackrock. Morgan Stanley's customer arbitration cases decided against them were 986 times higher than Blackrock's. In a similar conclusion, USB had 1322 misconduct regulatory fines, which was 188 times higher than Blackrock, while UBS had 614 customer arbitration legal proceedings decided against them, which was 614 times higher than Blackrock. A research question (RQ) arises from our study: Why are misconduct and customer arbitration cases so much higher for Morgan Stanley and UBS? These are substantial multinational global asset management firms, so their behavior potentially impacts a large population of retail and institutional investors. This RQ should be investigated in the future, perhaps using the single or contrasting case study method. Another angle to that RQ would be to examine the impacts of Morgan Stanley and UBS's unethical behavior pursuant to the GRI 206-1 anti-corruption governance factor of the ESG construct, such as retail and institutional investors. Our study did not address the customer side of the equation, so other researchers should pursue that.

Finally, we recommend that future researchers expand this type of analysis beyond the financial industry to manufacturing and all other sectors, using combinations of the methods outlined above. Those studies should randomly sample regions around the world beyond the USA.

**Author Contributions:** Conceptualization, K.D.S.; methodology, K.D.S.; statistical software (Python version 3.12), K.D.S.; formal analysis, K.D.S.; investigation, K.D.S.; Literature review, N.R.V. and K.D.S.; data collection, K.D.S.; writing—original draft, K.D.S.; writing—review and editing, K.D.S. and N.R.V.; writing—response to viewers, N.R.V. and K.D.S.; visualization, K.D.S.; project management, K.D.S. All authors have read and agreed to the published version of the manuscript.

**Funding:** The researchers received no external funding.

**Institutional Review Board Statement:** IRB was not needed for this study as the data accessed as not related to human subjects review and data was collected from publicly available databases.

**Informed Consent Statement:** Public data was collected for this study, so informed consent was irrelevant.

**Data Availability Statement:** Authors will provide data from Harvard University.

**Conflicts of Interest:** The authors declare no conflicts of interest or monetary benefit from this study and disclose that solely as a matter of employment they have had common place transactions or disputes concerning mandatory pension fund investments with asset management firms in this population which had no bearing on this study.

## Appendix A

**Table A1.** Evidence data collected from SEC, FINRA, NASD and U.S. Courts.

| Firm | Firm Misconduct | Arbitration Decided Against Firm |
|---|---|---|
| BlackRock Inc | https://files.brokercheck.finra.org/firm/firm_38642.pdf, accessed on 15 October 2024 | https://www.finra.org/arbitration-mediation/arbitration-awards-online?aao_radios=all&search=%22respondent(s):+blackrock%22&field_case_id_text=&field_core_official_dt[min]=&field_core_official_dt[max]=&field_document_type_tax[4224]=4224&field_forum_tax=All&field_special_case_type_tax=All, accessed on 15 October 2024 |
| Vanguard Group | https://files.brokercheck.finra.org/firm/firm_22081.pdf, accessed on 15 October 2024 | https://www.finra.org/arbitration-mediation/arbitration-awards-online?aao_radios=all&search=%22respondent(s):+vanguard%22&field_case_id_text=&field_core_official_dt[min]=&field_core_official_dt[max]=&field_document_type_tax[4224]=4224&field_forum_tax=All&field_special_case_type_tax=All, accessed on 15 October 2024 |
| Fidelity Investments | https://files.brokercheck.finra.org/firm/firm_17507.pdf, accessed on 15 October 2024 | https://www.finra.org/arbitration-mediation/arbitration-awards-online?aao_radios=all&search=%22respondent(s):+fidelity%22&field_case_id_text=&field_core_official_dt[min]=&field_core_official_dt[max]=&field_document_type_tax[4224]=4224&field_forum_tax=All&field_special_case_type_tax=All, accessed on 15 October 2024 |
| State Street Corporation | https://files.brokercheck.finra.org/firm/firm_285852.pdf, accessed on 15 October 2024 | https://www.finra.org/arbitration-mediation/arbitration-awards-online?aao_radios=all&search=%22respondent(s):+state+street%22&field_case_id_text=&field_core_official_dt[min]=&field_core_official_dt[max]=&field_document_type_tax[4224]=4224&field_forum_tax=All&field_special_case_type_tax=All, accessed on 15 October 2024 |
| Morgan Stanley | https://brokercheck.finra.org/firm/summary/149777, accessed on 15 October 2024 | https://www.finra.org/arbitration-mediation/arbitration-awards-online?aao_radios=all&search=%22respondent(s):+morgan+stanley%22&field_case_id_text=&field_core_official_dt[min]=&field_core_official_dt[max]=&field_document_type_tax[4224]=4224&field_forum_tax=All&field_special_case_type_tax=All, accessed on 15 October 2024 |
| UBS | https://brokercheck.finra.org/firm/summary/7654, accessed on 15 October 2024 | https://www.finra.org/arbitration-mediation/arbitration-awards-online?aao_radios=all&search=%22respondent(s):+UBS%22&field_case_id_text=&field_core_official_dt[min]=&field_core_official_dt[max]=&field_document_type_tax[4224]=4224&field_forum_tax=All&field_special_case_type_tax=All, accessed on 15 October 2024 |
| J.P. Morgan Chase & Co | https://brokercheck.finra.org/firm/summary/18718, accessed on 15 October 2024 | https://www.finra.org/arbitration-mediation/arbitration-awards-online?aao_radios=all&search=%22respondent(s):+JP+Morgan+Chase%22&field_case_id_text=&field_core_official_dt[min]=&field_core_official_dt[max]=&field_document_type_tax[4224]=4224&field_forum_tax=All&field_special_case_type_tax=All, accessed on 15 October 2024 |
| Goldman Sachs | https://brokercheck.finra.org/firm/summary/361, accessed on 15 October 2024 | https://www.finra.org/arbitration-mediation/arbitration-awards-online?aao_radios=all&search=%22respondent(s):+Goldman+Sachs%22&field_case_id_text=&field_core_official_dt[min]=&field_core_official_dt[max]=&field_document_type_tax[4224]=4224&field_forum_tax=All&field_special_case_type_tax=All, accessed on 15 October 2024 |

**Table A1.** *Cont.*

| Firm | Firm Misconduct | Arbitration Decided Against Firm |
|------|----------------|-------------------------------|
| BNY Mellon | https://brokercheck.finra.org/firm/summary/47268, accessed on 15 October 2024 | https://www.finra.org/arbitration-mediation/arbitration-awards-online?aao_radios=all&search=%22respondent(s):+BNY+Mellon%22&field_case_id_text=&field_core_official_dt[min]=&field_core_official_dt[max]=&field_document_type_tax[4224]=4224&field_forum_tax=All&field_special_case_type_tax=All, accessed on 15 October 2024 |
| Invesco | https://brokercheck.finra.org/firm/summary/289, accessed on 15 October 2024 | https://www.finra.org/arbitration-mediation/arbitration-awards-online?aao_radios=all&search=%22respondent(s):+Invesco%22&field_case_id_text=&field_core_official_dt[min]=&field_core_official_dt[max]=&field_document_type_tax[4224]=4224&field_forum_tax=All&field_special_case_type_tax=All |
| The Capital Group | https://adviserinfo.sec.gov/firm/summary/129615, accessed on 15 October 2024 | uncertain due to various subsidiary names |
| Franklin Group Financial Resources | https://brokercheck.finra.org/firm/summary/6627, accessed on 15 October 2024 | https://www.finra.org/arbitration-mediation/arbitration-awards-online?aao_radios=all&search=%22Franklin+Group%22&field_case_id_text=&field_core_official_dt[min]=&field_core_official_dt[max]=&field_forum_tax=All&field_special_case_type_tax=All, accessed on 15 October 2024 |
| PGIM (Fixed Income) | https://adviserinfo.sec.gov/firm/summary/105676, accessed on 15 October 2024 | uncertain due to various subsidiary names |
| Northern Trust | https://brokercheck.finra.org/firm/summary/7927, accessed on 15 October 2024 | https://www.finra.org/arbitration-mediation/arbitration-awards-online?aao_radios=all&search=%22respondent(s):+Northern+Trust%22&field_case_id_text=&field_core_official_dt[min]=&field_core_official_dt[max]=&field_document_type_tax[4224]=4224&field_forum_tax=All&field_special_case_type_tax=All, accessed on 15 October 2024 |
| Wellington Management | https://brokercheck.finra.org/firm/summary/7536, accessed on 15 October 2024 | uncertain due to various subsidiary names |
| TIAA-CREF | https://brokercheck.finra.org/firm/summary/20472, accessed on 15 October 2024 | https://www.finra.org/arbitration-mediation/arbitration-awards-online?aao_radios=all&search=%22respondent(s):+TIAA-CREF%22&field_case_id_text=&field_core_official_dt[min]=&field_core_official_dt[max]=&field_document_type_tax[4224]=4224&field_forum_tax=All&field_special_case_type_tax=All, accessed on 15 October 2024 |

Explanation for websites: For the sources listed as websites in the appendix, the data was compiled from reputable and authoritative sources to ensure accuracy and reliability. These sources include:

- Government Websites: Data from government websites was utilized for foundational information, regulatory guidance, and industry statistics. This includes datasets and reports made publicly available by federal and state entities.
- Securities and Exchange Commission (SEC): The SEC website provided essential data related to regulatory compliance, market activities, and corporate filings. This resource was instrumental in accessing publicly available information on financial markets and ensuring alignment with regulatory standards.
- Financial Industry Regulatory Authority (FINRA): FINRA's website served as a valuable resource for information on brokerage services, market rules, and investor protection policies. FINRA data ensured that insights into market regulations and compliance were current and comprehensive.

Each of these sources was selected for its credibility and direct relevance to the research topic. The information obtained reflects the most recent updates and standards set forth by these agencies.

**Table A2.** Data Collection Sources and Disclosure Records for Selected Financial Institutions.

| Company Name | Number of Additional Disclosures | | | | Data Collection Source Links | | |
|---|---|---|---|---|---|---|---|
| Sources: | https://www.sustainalytics.com/esg-rating/blackrock-inc/1008281659, accessed on 15 October 2024 | | | | https://www.csrhub.com/CSR_and_sustainability_information/BlackRock-Inc, accessed on 15 October 2024 | | |
| Note: State Stree Capital had two more disclosure records all totaled 34 | | | | | https://brokercheck.finra.org/firm/summary/30107, accessed on 15 October 2024 | https://files.brokercheck.finra.org/firm/firm_285852.pdf, accessed on 15 October 2024 | |
| Morgan Stanley had six additional disclosure records: | | | https://brokercheck.finra.org/firm/summary/7556, accessed on 15 October 2024 | https://brokercheck.finra.org/firm/summary/8209, accessed on 15 October 2024 | https://brokercheck.finra.org/firm/summary/30992, accessed on 15 October 2024 | https://brokercheck.finra.org/firm/summary/34925, accessed on 15 October 2024 | https://brokercheck.finra.org/firm/summary/29106, accessed on 15 October 2024 |
| E*Trade from Morgan Stanley had 93 regulatory fines, 137 arbitration decisions, and 50 non-registered disclosures | | | | | | | https://files.brokercheck.finra.org/firm/firm_29106.pdf, accessed on 15 October 2024 |
| Morgan Stanley's Horace Mann was another entity with 6 more disclosures | | | | | https://brokercheck.finra.org/firm/summary/11643, accessed on 15 October 2024 | | |
| UBS had additional disclosure records | https://brokercheck.finra.org/firm/summary/107726, accessed on 15 October 2024 | https://brokercheck.finra.org/firm/summary/2692, accessed on 15 October 2024 | https://brokercheck.finra.org/firm/summary/8174, accessed on 15 October 2024 | https://brokercheck.finra.org/firm/summary/583, accessed on 15 October 2024 | https://brokercheck.finra.org/firm/summary/44883, accessed on 15 October 2024 | https://brokercheck.finra.org/firm/summary/13042, accessed on 15 October 2024 | |
| J.P. Morgan Chase & Co had several disclosure records | | https://brokercheck.finra.org/firm/summary/79, accessed on 15 October 2024 | https://brokercheck.finra.org/firm/summary/17116, accessed on 15 October 2024 | https://brokercheck.finra.org/firm/summary/20989, accessed on 15 October 2024 | https://brokercheck.finra.org/firm/summary/28432, accessed on 15 October 2024 | https://brokercheck.finra.org/firm/summary/15733, accessed on 15 October 2024 | https://brokercheck.finra.org/firm/summary/1326, accessed on 15 October 2024 |
| Goldman Sachs Australia was closed but had 1 record | | https://brokercheck.finra.org/firm/summary/10117, accessed on 15 October 2024 | https://brokercheck.finra.org/firm/summary/48015, accessed on 15 October 2024 | https://brokercheck.finra.org/firm/summary/3466, accessed on 15 October 2024 | | | |
| BNY Mellon had several records | | https://brokercheck.finra.org/firm/summary/17454, accessed on 15 October 2024 | https://brokercheck.finra.org/firm/summary/231, accessed on 15 October 2024 | | | | |
| Investico had several records | | https://brokercheck.finra.org/firm/summary/7369, accessed on 15 October 2024 | https://brokercheck.finra.org/firm/summary/6939, accessed on 15 October 2024 | https://brokercheck.finra.org/firm/summary/14007, accessed on 15 October 2024 | | | |

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
