# Peer review of "Evaluating the Anti-Corruption Factor in Environmental, Social, and Governance Indices by Sampling Large Financial Asset Management Firms"

_sustainability, doi:10.3390/su162310240_

Round 1
Reviewer 1 Report
Comments and Suggestions for Authors
Ultimately this study failed to capture my attention in a useful manner. The framing of the study is somewhat sensationalist in places, more so than is deserving of an academic study, for instance the claim in the abstract that faulty ESG data could cause or economies to fail is simply not grounded in evidence or proper reasoning. Similar sensational framing is used elsewhere to make the article sound attractive, but the truth is the article makes an already widely known point, that ESG data is incomplete and often not reported consistently for a variety or reasons.
The connection to corruption is proxied by number of legal disputes, but these are not equivalent concepts. Legal challenges can and arguably more often than not will arise from a lack of duty (negligence in the severe case) but far less likely from deliberate corrupt actions.
The study associates the absence of quality corruption information in some mainstream ESG indicators/frameworks as a means to contest the quality of ESG data, yet it is not the purpose of such data to highlight such information. This is important in two regards (i) the purpose of ESG is to very performance in relation to environment, social attributes and core governance procedures. At best corruption will feature as one aspect of the latter. (ii) ESG data providers offer controversies variables designed to capture information such as corruption. The present study makes no reference to such data and this is a massive oversight in my opinion. Controversies scores are increasingly widely used to weight final ESG scores, and hence data regarding corruption might be implicitly embedded in the ESG data analysed.
I appreciate the effort that the authors have invested into the methodology, but am of the view that the findings are largely established by previous literature, and that the presentation is too bold and in some cases overly confident in using unsupported and contestable claims in order to keep the reader engaged.
I do not see a pathway through to publication without such extensive change that it ought then be considered a different submission.
Comments on the Quality of English LanguageSee comments above about the objective language (or lack of in places) used.
Author Response
Reviewer 1 Reviewer Comments Response from the Authors
Ultimately this study failed to capture my attention in a useful manner. The framing of the study is somewhat sensationalist in places, more so than is deserving of an academic study, for instance the claim in the abstract that faulty ESG data could cause or economies to fail is simply not grounded in evidence or proper reasoning. Similar sensational framing is used elsewhere to make the article sound attractive, but the truth is the article makes an already widely known point, that ESG data is incomplete and often not reported consistently for a variety or reasons.
Thank you to this reviewer for their effort in doing this peer review. We acknowledge that the reviewer found some of our claims, particularly in the abstract (and also the conclusions), used strong pessimistic phrasing, namely where we claimed that economies could be dramatically impacted due to some of the large $1 trillion asset management firms failing to report numerous legal cases decided against them. That was a bit far-fetched! The reviewer is correct in that academics ought to take a more objective perspective, which we have done in our revisions (we hope this is much improved). We did want to emphasize at least to the review, the importance of inaccurate ESG scores, even though it may seem obvious. We modified our abstract and conclusions to tame down our claims to match reality, namely we used this statement: "The problem is that current Environment Social Governance (ESG) indexes are flawed because the data is incomplete, they are often not reported consistently, and some factors may be irrelevant to what firms produce. Some regulators in the financial services industry emphasize reporting CO2 emissions (environment factor) when rented offices and the internet are the key resources leveraged for production, rather than disclosing governance issues like money laundering, corruption, and unethical behavior." We feel we discovered something new, what was not known by anyone except the insiders at the asset management firms, and it was unreported in the literature, being that we found numerous unreported legal disputes decided against a few large asset management firms (over 1000 cases for one firm in our sample) - we use the analogy that when new ships were launched into space, we already knew much of what was going to be discovered, yet we always learned something new from each voyage, and that is the goal of science, to discover new knowledge - discovery is valuable in and of itself. In our study, what we discovered will be more than novel; it will have significant financial value to stakeholders and shareholders of those firms (the population) if other researchers extend our findings - we refer to the VW case (in the auto industry) as an example of the magnitude of impact to their shareholders , likewise, we feel there will be value from our ESG discovery to the financial shareholders. We also added more interpretations to section 4.2 as well as the conclusions - here is a quote: "Fourth, when considering our post-hoc root cause analysis from the stakeholder theory and public interest theory perspectives, based on interprepting the rigorous Bayesian statistical results against GRI 206-1 anti-corruption governance factor of the ESG construct, it was clear that one firm was a good performer and two firms were significantly bad performers. Blackrock clearly was the best performer in the sample. Morgan Stanley was the worst performer, with 1758 misconduct regulatory fines, which was 258 times higher than the best performer Blackrock, and Morgan Stanley’s customer arbitration cases decided against them was 986, which was 986 times higher than Blackrock. In a similar conclusion, USB has 1322 misconduct regulatory fines which was 188 times higher than Blackrock, while UBS had 614 customer arbitration legal proceedings decided against them, which was 614 times higher than Blackrock. A research question (RQ) arised from our study: Why are misconducts and customer arbitration cases so much higher for Morgan Stanley and UBS? These are huge multinational global asset management firms so their behavior potential impacts a large population of retail and institutional investors. This RQ ought to be investigated in the future perhaps using the single or contrasting case study method. another angle to that RQ would be to investigate the impacts of Morgan Stanley and UBS unethical behavior pursuant to the GRI 206-1 anti-corruption governance factor of the ESG construct, such as retail and institutional investors. Our study did not address the customer side of the equation, so other researchers ought to pursue that. "
The connection to corruption is proxied by number of legal disputes, but these are not equivalent concepts. Legal challenges can and arguably more often than not will arise from a lack of duty (negligence in the severe case) but far less likely from deliberate corrupt actions.
Again that you for this. We cited the Global Reporting Institute (GRI) standard number GR100 as the explicit link between corruption, legal disputes and the 'G' factor in the ESG construct. GR100 specifically addresses the number of legal disputes reported where the firm was named (or unethically not reported). We included a quote from the GRI in our original version. Please confirm if the reviewer wants us to add more explanation or remove this claim. Note we appreciated what the reviewer implied in the last sentence, and we added a paraphrase of that into the paper, namely that it is possible that some firms forgot to report legal disputes out of excusable negligence (forgetting to report legal disputes), rather than unethically choosing to deliberately not report legal disputes decided against the firm, especially where there have been many years or decades since the firm had any reportable disputes - keep in mind though that the two of the sampled asset management firms we found involved over 1000 unreported legal cases decided against them, one had a case where the firm had to pay over $5 million in regulatory fines for their misconduct behavior.
The study associates the absence of quality corruption information in some mainstream ESG indicators/frameworks as a means to contest the quality of ESG data, yet it is not the purpose of such data to highlight such information. This is important in two regards (i) the purpose of ESG is to very performance in relation to environment, social attributes and core governance procedures. At best corruption will feature as one aspect of the latter. (ii) ESG data providers offer controversies variables designed to capture information such as corruption. The present study makes no reference to such data and this is a massive oversight in my opinion. Controversies scores are increasingly widely used to weight final ESG scores, and hence data regarding corruption might be implicitly embedded in the ESG data analyzed.
Thank you to the reviewer. This was an insightful point which we may not have emphasized in our paper. We concur that ESG indexes have been produced primarily to compare a firm's compliance to a subset of the United Nations SDG, being ESG, and subsequently only a subset of ESG (the 'G' factor in our study). We pointed out that our study addressed only one item in the ESG construct. We felt we should add that to our abstract to make that clear at the outset, to condition our findings, while novel, they are only a small impact to all the numerous items examined within the ESG construct of the Global Reporting Institute ESG framework. We also concur that controversies are being utilized as the sub-factor to measure and report corrurption as well as legal disputes where the firm is named as a defendant - we mentioned that in our conclusion but we felt it was such a good point that we added a mention of that to our abstract. We hope we captured your idea but let us know if this needs more work.
I appreciate the effort that the authors have invested into the methodology, but am of the view that the findings are largely established by previous literature, and that the presentation is too bold and in some cases overly confident in using unsupported and contestable claims in order to keep the reader engaged.
Thank you - yes we feel we designed and applied a very rigourous research and statistical testing framework. We also feel that we discovered something new, something that was not known by other researchers and the scientific community, except the insiders at the asset management firms, and it was unreported in the literature, being that we found numerous unreported legal disputes decided against a few large asset management firms (over 1000 cases for one firm in our sample) - we use the analogy that when new ships were launched into space, we already knew much of what was going to be discovered, yet we always learned something new from each voyage, and that is the goal of science, to discover new knowledge - discovery is valuable in and of itself. In our study, what we discovered will be more than novel; it will have significant financial value to stakeholders and shareholders of those firms (the population) if other researchers extend our findings. We argue that our claims were substantiated - we proved there were thousands of unreported legal cases against a few of the large asset management firms in our sample. We are unsure how much more evidence we would have to include - please could you describe what additional evidence the reviewer would like to see to substantiate our hypotheses?
I do not see a pathway through to publication without such extensive change that it ought then be considered a different submission.
We hope the reviewer will determine there is value in publishing our revised manuscript since it used a rigorous research/statistical framework, novel techniques (artificial intelligence) to collect data not reported elsewhere, and we found the data supported our hypotheses. We even included some of the data in our appendix since we concur this was so novel that other researchers may doubt that the data existed (albeit well-concealed) in the public domain of the government. We also made changes to the paper to address the requirements of the other four reviewers so as a whole we feel the paper has improved for you.
Reviewer 2 Report
Comments and Suggestions for Authors
Well developed research frame but need to add more theoretical support.
In addition, this study needs to explain why the evaluating ESG Scores are developed for industry in introduction part.
Last, current study needs to demonstrate the contribution of present research as well as the practical founding based on previous researches.
Comments on the Quality of English LanguageWell written english, but need to verify the gramma in manuscript.
Author Response
Reviewer 2 Reviewer Comments Response from the Authors
Well developed research frame but need to add more theoretical support.
We have explanations for the research frame, namely how we developed the ESG three-factor framework based on the CSRhub construct (it has 4 factors). We also discussed how we used random sampling based on a third part indepdent study, to select the 20 firms, then to purposively reduce the sample to 14 firms based on if the ESG data were publically available. We also explained how and why we used certain parametric statistical techniques including Bayesian correlation and bootstrapping. Citations of key elements were given to recognized studies in the literature (over 7 citation just in that section). We can add more if the review wishes but we would be adding to the already long page count.
In addition, this study needs to explain why the evaluating ESG Scores are developed for industry in introduction part.
Good point. We explained how regulators in the financial services industry incorrectly emphasize reporting CO2 emissions (environment factor) yet the key resources leveraged for production by asset management firms are rented offices and the internet [not mining or other agriculture activities with CO2 emissions] - we argued that governance issues like money laundering, corruption, and unethical behavior would be more relevant. We explained how incorrect ESG scores impact stakeholders and shareholder, the implications for readers and decision makers.
Last, current study needs to demonstrate the contribution of present research as well as the practical founding based on previous researches.
Thank you. We have reworked the abstract and conclusions to emphasize this - here is the abstract for example: "The problem is that current Environment Social Governance (ESG) indexes are flawed because the data is incomplete, not reported consistently, and some measured factors may be irrelevant to the industry. Regulators in the financial services industry emphasize reporting CO2 emissions (environment factor) yet the key resources leveraged for production are rented offices and the internet - governance issues like money laundering, corruption, and unethical behavior would be more relevant. To investigate this problem, we sampled the biggest finance and insurance industry firms in the USA having the greatest economic impact, those managing at least $1 trillion in assets. We used artificial intelligence to collect data about undisclosed legal decisions against firms to measure the anti-corruption governance factor GRI 206-1 defined by the Global Reporting Institute for measuring global sustainable development goals (SDGs) in support of the United Nations’ SDGs and ESG. We applied Bayesian correlation with bootstrapping to test our hypotheses, followed by root cause analysis. We found ESG ratings from providers did not reflect legal cases decided against firms; the Bayesian BF₊₀ odds ratio was 3005 (99% confidence intervals were 0.617, 0.965), misconduct fines and arbitration legal case counts were significantly related for the same firm (Vovk-Selke maximum p-ratio was 4411), but most ESG scores were significantly different for the same firm. We found three other studies in the literature which corroborated some of our findings that specific firms in our sample were considered unethical. We proposed implications based on the Theory of Public Interest and Stakeholder Theory. "
Reviewer 3 Report
Comments and Suggestions for Authors
I like the paper. The topic is current and important nowadays and the authors proposed an ambitious objective.
The paper investtigated the ESG scores and how ESG scores from several providers were related considering the fact that the same firm can have different ESG score. Also, the paper analyzed the impact of anti-corruption human-rights governance on ESG scores.
The results were corelated with the results obatained in other studies, and in the same time, the authors argued their results.
The authors highlighted the limitations of their study and also proposed recommendations and future research which can confirm their results.
It is a good paper and my recommendation is to be accepted for publication based on some reasons, such as: a very interesting topic, good data collected, use of an adequate methodology, contribution to the literature in the field, and also the relevance of the conclusions. However, next you may find some comments.
From my point of view, the Abstract is too long and need to be resumed to the objective of the paper, methodology employed and results. The other information can be offered in Introduction.
Author Response
Reviewer 3
Reviewer Comments Response from the Authors
I like the paper. The topic is current and important nowadays and the authors proposed an ambitious objective.
Thank you. We put a lot of effort into aligning our study with the ESG special issue - you can see our references are very new (most were 2023-2024) and our data were collected in the middle of 2024 so it is very current.
The paper investigated the ESG scores and how ESG scores from several providers were related because the same firm can have different ESG score. Also, the paper analyzed the impact of anti-corruption human-rights governance on ESG scores.
Thank you. Yes we actually focused more on the anti-corruption dimension within the GRI 206-1 ESG factor. Here is some of the interpretations we added to emphasize that. Fourth, when considering our post-hoc root cause analysis from the stakeholder theory and public interest theory perspectives, based on interprepting the rigorous Bayesian statistical results against GRI 206-1 anti-corruption governance factor of the ESG construct, it was clear that one firm was a good performer and two firms were significantly bad performers. Blackrock clearly was the best performer in the sample. Morgan Stanley was the worst performer, with 1758 misconduct regulatory fines, which was 258 times higher than the best performer Blackrock, and Morgan Stanley’s customer arbitration cases decided against them was 986, which was 986 times higher than Blackrock. In a similar conclusion, USB has 1322 misconduct regulatory fines which was 188 times higher than Blackrock, while UBS had 614 customer arbitration legal proceedings decided against them, which was 614 times higher than Blackrock. A research question (RQ) arised from our study: Why are misconducts and customer arbitration cases so much higher for Morgan Stanley and UBS? These are huge multinational global asset management firms so their behavior potential impacts a large population of retail and institutional investors. This RQ ought to be investigated in the future perhaps using the single or contrasting case study method. another angle to that RQ would be to investigate the impacts of Morgan Stanley and UBS unethical behavior pursuant to the GRI 206-1 anti-corruption governance factor of the ESG construct, such as retail and institutional investors. Our study did not address the customer side of the equation, so other researchers ought to pursue that.
The results were corelated with the results obtained in other studies, and in the same time, the authors argued their results. Thank you.
The authors highlighted the limitations of their study and also proposed recommendations and future research which can confirm their results.
Thank you. Yes we also added more to the recommendations for additional research. Here is what we added: "Fourth, when considering our post-hoc root cause analysis from the stakeholder theory and public interest theory perspectives, based on interprepting the rigorous Bayesian statistical results against GRI 206-1 anti-corruption governance factor of the ESG construct, it was clear that one firm was a good performer and two firms were significantly bad performers. Blackrock clearly was the best performer in the sample. Morgan Stanley was the worst performer, with 1758 misconduct regulatory fines, which was 258 times higher than the best performer Blackrock, and Morgan Stanley’s customer arbitration cases decided against them was 986, which was 986 times higher than Blackrock. In a similar conclusion, USB has 1322 misconduct regulatory fines which was 188 times higher than Blackrock, while UBS had 614 customer arbitration legal proceedings decided against them, which was 614 times higher than Blackrock. A research question (RQ) arised from our study: Why are misconducts and customer arbitration cases so much higher for Morgan Stanley and UBS? These are huge multinational global asset management firms so their behavior potential impacts a large population of retail and institutional investors. This RQ ought to be investigated in the future perhaps using the single or contrasting case study method. another angle to that RQ would be to investigate the impacts of Morgan Stanley and UBS unethical behavior pursuant to the GRI 206-1 anti-corruption governance factor of the ESG construct, such as retail and institutional investors. Our study did not address the customer side of the equation, so other researchers ought to pursue that. "
It is a good paper and my recommendation is to be accepted for publication based on some reasons, such as: a very interesting topic, good data collected, use of an adequate methodology, contribution to the literature in the field, and also the relevance of the conclusions. However, next you may find some comments. Thank you.
From my point of view, the Abstract is too long and need to be resumed to the objective of the paper, methodology employed and results. The other information can be offered in Introduction.
Thank you. We have reworked the abstract as advised. We reduced the abstract word count to 248 and revised to emphasize the findings with effect sizes - here it is: "The problem is that current Environment Social Governance (ESG) indexes are flawed because the data is incomplete, not reported consistently, and some measured factors may be irrelevant to the industry. Regulators in the financial services industry emphasize reporting CO2 emissions (environment factor) yet the key resources leveraged for production are rented offices and the internet - governance issues like money laundering, corruption, and unethical behavior would be more relevant. To investigate this problem, we sampled the biggest finance and insurance industry firms in the USA having the greatest economic impact, those managing at least $1 trillion in assets. We used artificial intelligence to collect data about undisclosed legal decisions against firms to measure the anti-corruption governance factor GRI 206-1 defined by the Global Reporting Institute for measuring global sustainable development goals (SDGs) in support of the United Nations’ SDGs and ESG. We applied Bayesian correlation with bootstrapping to test our hypotheses, followed by root cause analysis. We found ESG ratings from providers did not reflect legal cases decided against firms; the Bayesian BF₊₀ odds ratio was 3005 (99% confidence intervals were 0.617, 0.965), misconduct fines and arbitration legal case counts were significantly related for the same firm (Vovk-Selke maximum p-ratio was 4411), but most ESG scores were significantly different for the same firm. We found three other studies in the literature which corroborated some of our findings that specific firms in our sample were considered unethical. We proposed implications based on the Theory of Public Interest and Stakeholder Theory. "
Reviewer 4 Report
Comments and Suggestions for Authors
History has repeatedly demonstrated that the out-of-control financial industry devastates entire sectors of the global economy, affecting the objectives of sustainable development. The authors of the article formulate a research theme very strongly anchored in economic and social realities. The starting point of the research is as correct as possible. "A financial services firm ought to have more impact on governance factors such as ethics, money laundering, and anticorruption.", because "Good governance ensures that companies comply with relevant laws and regulations, including those related to environmental protection and social equity, which is necessary for sustainable business operations and contributes to achieving the SDGs."
We consider the reservations regarding the reliability of the ESG ratings to be well documented and that the scores do not reflect the wrong conduct of the large financial asset management companies.
We draw attention to a structural imbalance in paragraph 4.2 Stakeholder theory point of view. In our opinion, this paragraph must be approached together with the theory of public interest, in the sense that not only the stakeholders must win, but also the public interest in general by observing the norms of social coexistence. We can agree "If the firm has more than sufficient profit to pay for arbitration as well as misconduct fines, penalties, sanctions, and legal costs, this may be considered as a factor of production, a necessary operating expense.", but this expense should not be considered normal nor tax deductible, being taxed as such and presented as a vulnerability of the company.
Author Response
Reviewer 4 Reviewer Comments Response from the Authors
History has repeatedly demonstrated that the out-of-control financial industry devastates entire sectors of the global economy, affecting the objectives of sustainable development. The authors of the article formulate a research theme very strongly anchored in economic and social realities. The starting point of the research is as correct as possible. "A financial services firm ought to have more impact on governance factors such as ethics, money laundering, and anticorruption.", because "Good governance ensures that companies comply with relevant laws and regulations, including those related to environmental protection and social equity, which is necessary for sustainable business operations and contributes to achieving the SDGs."
Thank you. Exactly we concur. To illustrate that, we added more to cover the Public Interest Theory perspective to balance with Stakeholder Theory in section 4.2. Here is ome of what we added: "There is usually another point of view on any findings, and we discuss two opposing theories here. Stakeholder theory suggests that firms exist to serve their shareholders and that means to maximize profits which will be distributed to shareholders as dividends [26]. Under the stakeholder theory perspective, firms may be thought of as acting within their mission to avoid disclosing misconduct and arbitration decisions decided against them. Certainly, one could imagine if UBS and Morgan Stanley listed all of their misconduct and arbitration legal proceedings decided against them in their prospectus, annual report, and other marketing brochures, the result would not be favorable for their shareholders. Yet another perspective to stakeholder theory in this study is that some firms may have a well-thought-out strategy to diminish the seriousness of misconduct and arbitration legal proceedings decided against them. If the firm has more than sufficient profit to pay for arbitration as well as misconduct fines, penalties, sanctions, and legal costs, this may be considered as a factor of production, a necessary operating expense. We noted that Park and Jang [70] adopted this perspective as they explored how companies could develop strategies to comply with ESG. We are not trying to justify this behavior, but rather, we are trying to rationalize why this unethical behavior continues to occur. It was clear from the data collection that the misconduct and arbitration legal proceeding decisions dated back decades. This is not a post-COVID phenomenon or a new regulatory initiative. it appears this is part of the production formula, a standard operating procedure. It is certainly an interesting topic to research in the future. A contrary perspective to stakeholder theory is based on the theory of public interest. Public interest theory was found by Taylor and Whitton [71] to be ambiguous and pluralistic. In general, Taylor and Whitton [71] determined the best explanation for the theory of public interest was to meet the common objectives valued by the community, a fair and proper balance between individual and group interests, ensuring the delivery of democratic expectations. If we apply public interest theory to the findings of this study, we must reject our stakeholder theory proposition discussed above. Our argument in favor of public interest theory is if only the firm and its shareholders benefit from unethical disclosures for ESG ratings, that will be unfair to other firms who are honest, unfair to group investors such as pension plan managers, and unfair to individual retail investors who would not be award of the higher risk associated with firms who do not disclose legal cases decided against them in their ESG scores. Whether the affected parties are non-shareholders is not the point - lack of disclosure and lack of transparency are contrary to public interest theory. From the public interest theory standpoint, we could also assert that disclosing fines paid and legal cases decided against a firm would be relevant to report for an ESG score due to social coexistence and social norms in our understand of global culture. While above we agued in favor of stakeholder theory, that a firm can consider misconduct fines, penalties, sanctions, and legal cost as a proprietary factor of production which need not be disclosed, we could contrast to that, the need for society in general to absorb the losses for those negatively impacted by such nondisclosures. We could go so far as to juxtapose public interest theory as a tax to be imposed on firms who leverage nontransparancy and do not disclose legal fines or cases against them. We mean that such firms, who other researchers have labeled unethical, ought to be taxed for this exception just as a company or country who pollutes the environment must either pay a fine, tax, or tariff in the form of carbon tax on exports or sales. Subsequently, in an accounting sense, such legal nondisclosures, and potentially low ESG scores, could be considered non tax deductible in the balance sheet, and it ought to be presented to shareholders and stakeholder as as a vulnerability of the company. Given that stakeholder theory states that firms should act in the best interests of shareholders, we find it difficult to understand that a lay person, a shareholder or retail investor, would not benefit from knowing about the frequency of misconduct and arbitration legal proceedings decided against the firm."
We consider the reservations regarding the reliability of the ESG ratings to be well documented and that the scores do not reflect the wrong conduct of the large financial asset management companies.
Thank you. To expland on your idea we added more to section 4.2 and the conclusions. Here is a quote to illustrate that: "Fourth, when considering our post-hoc root cause analysis from the stakeholder theory and public interest theory perspectives, based on interprepting the rigorous Bayesian statistical results against GRI 206-1 anti-corruption governance factor of the ESG construct, it was clear that one firm was a good performer and two firms were significantly bad performers. Blackrock clearly was the best performer in the sample. Morgan Stanley was the worst performer, with 1758 misconduct regulatory fines, which was 258 times higher than the best performer Blackrock, and Morgan Stanley’s customer arbitration cases decided against them was 986, which was 986 times higher than Blackrock. In a similar conclusion, USB has 1322 misconduct regulatory fines which was 188 times higher than Blackrock, while UBS had 614 customer arbitration legal proceedings decided against them, which was 614 times higher than Blackrock. A research question (RQ) arised from our study: Why are misconducts and customer arbitration cases so much higher for Morgan Stanley and UBS? These are huge multinational global asset management firms so their behavior potential impacts a large population of retail and institutional investors. This RQ ought to be investigated in the future perhaps using the single or contrasting case study method. another angle to that RQ would be to investigate the impacts of Morgan Stanley and UBS unethical behavior pursuant to the GRI 206-1 anti-corruption governance factor of the ESG construct, such as retail and institutional investors. Our study did not address the customer side of the equation, so other researchers ought to pursue that. "
We draw attention to a structural imbalance in paragraph 4.2 Stakeholder theory point of view. In our opinion, this paragraph must be approached together with the theory of public interest, in the sense that not only the stakeholders must win, but also the public interest in general by observing the norms of social coexistence. We can agree "If the firm has more than sufficient profit to pay for arbitration as well as misconduct fines, penalties, sanctions, and legal costs, this may be considered as a factor of production, a necessary operating expense.", but this expense should not be considered normal nor tax deductible, being taxed as such and presented as a vulnerability of the company.
Salient point. This was so relevant that we incorporated what the reviewer stated into our conclusions. Please have a look and let us know if we addressed this correctly. For your concenience here is what we stated: "There is usually another point of view on any findings, and we discuss two opposing theories here. Stakeholder theory suggests that firms exist to serve their shareholders and that means to maximize profits which will be distributed to shareholders as dividends [26]. Under the stakeholder theory perspective, firms may be thought of as acting within their mission to avoid disclosing misconduct and arbitration decisions decided against them. Certainly, one could imagine if UBS and Morgan Stanley listed all of their misconduct and arbitration legal proceedings decided against them in their prospectus, annual report, and other marketing brochures, the result would not be favorable for their shareholders. Yet another perspective to stakeholder theory in this study is that some firms may have a well-thought-out strategy to diminish the seriousness of misconduct and arbitration legal proceedings decided against them. If the firm has more than sufficient profit to pay for arbitration as well as misconduct fines, penalties, sanctions, and legal costs, this may be considered as a factor of production, a necessary operating expense. We noted that Park and Jang [70] adopted this perspective as they explored how companies could develop strategies to comply with ESG. We are not trying to justify this behavior, but rather, we are trying to rationalize why this unethical behavior continues to occur. It was clear from the data collection that the misconduct and arbitration legal proceeding decisions dated back decades. This is not a post-COVID phenomenon or a new regulatory initiative. it appears this is part of the production formula, a standard operating procedure. It is certainly an interesting topic to research in the future. A contrary perspective to stakeholder theory is based on the theory of public interest. Public interest theory was found by Taylor and Whitton [71] to be ambiguous and pluralistic. In general, Taylor and Whitton [71] determined the best explanation for the theory of public interest was to meet the common objectives valued by the community, a fair and proper balance between individual and group interests, ensuring the delivery of democratic expectations. If we apply public interest theory to the findings of this study, we must reject our stakeholder theory proposition discussed above. Our argument in favor of public interest theory is if only the firm and its shareholders benefit from unethical disclosures for ESG ratings, that will be unfair to other firms who are honest, unfair to group investors such as pension plan managers, and unfair to individual retail investors who would not be award of the higher risk associated with firms who do not disclose legal cases decided against them in their ESG scores. Whether the affected parties are non-shareholders is not the point - lack of disclosure and lack of transparency are contrary to public interest theory. From the public interest theory standpoint, we could also assert that disclosing fines paid and legal cases decided against a firm would be relevant to report for an ESG score due to social coexistence and social norms in our understand of global culture. While above we agued in favor of stakeholder theory, that a firm can consider misconduct fines, penalties, sanctions, and legal cost as a proprietary factor of production which need not be disclosed, we could contrast to that, the need for society in general to absorb the losses for those negatively impacted by such nondisclosures. We could go so far as to juxtapose public interest theory as a tax to be imposed on firms who leverage nontransparancy and do not disclose legal fines or cases against them. We mean that such firms, who other researchers have labeled unethical, ought to be taxed for this exception just as a company or country who pollutes the environment must either pay a fine, tax, or tariff in the form of carbon tax on exports or sales. Subsequently, in an accounting sense, such legal nondisclosures, and potentially low ESG scores, could be considered non tax deductible in the balance sheet, and it ought to be presented to shareholders and stakeholder as as a vulnerability of the company. Given that stakeholder theory states that firms should act in the best interests of shareholders, we find it difficult to understand that a lay person, a shareholder or retail investor, would not benefit from knowing about the frequency of misconduct and arbitration legal proceedings decided against the firm" We also returned to that idea of yours in the conclusions where we called for more research. Here is what we added: "Fourth, when considering our post-hoc root cause analysis from the stakeholder theory and public interest theory perspectives, based on interprepting the rigorous Bayesian statistical results against GRI 206-1 anti-corruption governance factor of the ESG construct, it was clear that one firm was a good performer and two firms were significantly bad performers. Blackrock clearly was the best performer in the sample. Morgan Stanley was the worst performer, with 1758 misconduct regulatory fines, which was 258 times higher than the best performer Blackrock, and Morgan Stanley’s customer arbitration cases decided against them was 986, which was 986 times higher than Blackrock. In a similar conclusion, USB has 1322 misconduct regulatory fines which was 188 times higher than Blackrock, while UBS had 614 customer arbitration legal proceedings decided against them, which was 614 times higher than Blackrock. A research question (RQ) arised from our study: Why are misconducts and customer arbitration cases so much higher for Morgan Stanley and UBS? These are huge multinational global asset management firms so their behavior potential impacts a large population of retail and institutional investors. This RQ ought to be investigated in the future perhaps using the single or contrasting case study method. another angle to that RQ would be to investigate the impacts of Morgan Stanley and UBS unethical behavior pursuant to the GRI 206-1 anti-corruption governance factor of the ESG construct, such as retail and institutional investors. Our study did not address the customer side of the equation, so other researchers ought to pursue that. "
Reviewer 5 Report
Comments and Suggestions for Authors
This article is designed to evaluate the scores of Environment Social Governance (ESG) by examining anti-corruption governance in large financial asset management firms. The overall methodology and procedures were valid, but the ABSTRACT should be re-summarized as there are no summarized conclusions and proper significant implications. Furthermore, the ESG should not be presented in the title instead of Environment Social Governance (ESG).
Author Response
Reviewer 5 Reviewer Comments Response from the Authors This article is designed to evaluate the scores of Environment Social Governance (ESG) by examining anti-corruption governance in large financial asset management firms. The overall methodology and procedures were valid, but the ABSTRACT should be re-summarized as there are no summarized conclusions and proper significant implications. Furthermore, the ESG should not be presented in the title instead of Environment Social Governance (ESG).
Thank you. We have reworked the abstract as advised. The change in title has also been made as advised. We changed the title to: "Evaluating the Anti-Corruption Governance Factor in Environment Social Governance Indexes by Sampling Large Financial Asset Management Firms" We reduced the abstract word count to 248 and revised to emphasize the findings with effect sizes - here it is: "The problem is that current Environment Social Governance (ESG) indexes are flawed because the data is incomplete, not reported consistently, and some measured factors may be irrelevant to the industry. Regulators in the financial services industry emphasize reporting CO2 emissions (environment factor) yet the key resources leveraged for production are rented offices and the internet - governance issues like money laundering, corruption, and unethical behavior would be more relevant. To investigate this problem, we sampled the biggest finance and insurance industry firms in the USA having the greatest economic impact, those managing at least $1 trillion in assets. We used artificial intelligence to collect data about undisclosed legal decisions against firms to measure the anti-corruption governance factor GRI 206-1 defined by the Global Reporting Institute for measuring global sustainable development goals (SDGs) in support of the United Nations’ SDGs and ESG. We applied Bayesian correlation with bootstrapping to test our hypotheses, followed by root cause analysis. We found ESG ratings from providers did not reflect legal cases decided against firms; the Bayesian BF₊₀ odds ratio was 3005 (99% confidence intervals were 0.617, 0.965), misconduct fines and arbitration legal case counts were significantly related for the same firm (Vovk-Selke maximum p-ratio was 4411), but most ESG scores were significantly different for the same firm. We found three other studies in the literature which corroborated some of our findings that specific firms in our sample were considered unethical. We proposed implications based on the Theory of Public Interest and Stakeholder Theory. "